# Drivers of antimicrobial resistance in pig production systems of Uganda

Adrian Muwonge [1] ✉, Tadeo Kakooza[2], Paul C. D. Johnson [2], Lawrence Kisuule[3,4],
Michael Kimaanga[3,4], Clovice Kankya[3], Barend Mark de Clare Bronsvoort[1] & Tiziana Lembo[2]

Increasing protein demand in low- and middle-income countries may accelerate livestock
intensification, antibiotic overuse and antimicrobial resistance (AMR) risk. Here, we examined
Uganda's growing pig sector, tracking 70 farmers and their pigs in semi-intensive and free-range
systems for a year. We investigated AMR and AMR gene abundance of 668 *Escherichia coli*, *Klebsiella*
and DNA isolated from 877 faecal samples using diffusion disc-method and qPCR, respectively. Pigs
in semi-intensive systems were 2.2 times more likely to exhibit AMR and had higher *ermB* levels. AMR
in free-range farmers was twice that of pigs but still 1.4 times less likely than in semi-intensive systems.
AMR prevalence increased by 0.76% per month. Potential transmission events were more likely on
semi-intensive farms (OR = 3.16, 95% CI: 2.1–4.3, $P < 0.001$), especially when farmers had higher *tetQ*
levels than pigs; the reverse was true for *ermB*. Intensified urban pig production may elevate AMR
risks, underscoring the need for targeted interventions.

Antimicrobial use (AMU) in livestock, particularly antibacterial use (ABU), is inextricably linked to the emergence of antimicrobial resistance (AMR)[1] in bacteria which is a major global health threat. Low- and middle-income countries (LMICs) are at particular risk, especially in sub-Saharan Africa[2] where an estimated 75 per 100,000 deaths are directly associated with AMR[3]. Much of our current understanding of AMR and its control is primarily anthropocentric[3,4], as reflected in clinical data focus on human population and in resource allocation. However, recent studies indicate that the AMR burden in communities could be high and driven by factors including but not limited to livestock production-associated characteristics[5].

Livestock production is a critical component of global agri-food systems. However, these systems are undergoing transformation in response to rapid population growth, increasing incomes, urbanisation and migration resulting in increased demand for animal protein[6,7] produced on shrinking land resources amidst climate-mediated challenges[6]. Some of these factors have historically influenced shifts from extensive to intensive livestock production in high-income countries[8] and are, therefore, expected to lead to similar shifts in LMICs[9]. Such transformations are typically associated with increased stocking densities[10] and productivity at the risk of a higher zoonotic disease burden[10] hence greater AMU as prophylaxis, treatment, and growth promotion[5]. Indeed, in pig production, AMU is expected to increase 3-fold in Asia and Africa by 2030[11,12] but the impact of this AMU rise on countries with fast-growing pork industries[13] like Uganda remains unknown.

High levels of AMU in livestock are a concern in terms of the emergence of AMR and onwards transmission to humans through the food chain. Such risks are particularly high in LMIC livestock production systems where humans and livestock live in close proximity[2,14]. Anecdotal reports indicate considerable variation in AMU between peri-urban (henceforth semi'-intensive) and extensive, rural (henceforth free-range) farms in Uganda[15]. It is likely that these differences will only increase as these production systems transition from free-range to semi-intensive/intensive. Beyond such predictions[16], AMR dynamics are complex and are often driven by multi-dimensional factors other than AMU, operating at the pathogen, host and production system levels.

Here, focusing on World Health Organization (WHO) high-priority *Enterobacteriacae Escherichia coli* and *Klebsiella*[17,18] species at the human-pig interface of Uganda, we comprehensively examine, through repeated sampling of human-pig pairs, whether differences in AMR exist across bacterial, host and production systems. Our specific objectives were to (1) quantify and compare the rates of phenotypic and genotypic resistance to predominantly used antibiotics, including multi-drug resistance profiles at bacterial (*E. coli*/and *Klebsiella* spp.) and host (farmers/pigs) levels in semi-intensive and free-range production systems, (2) identify the factors associated with this resistance and (3) investigate the potential for cross-species transmission at different spatial scales (farm, within and between production systems).

[1]The Digital One Health Laboratory at The Roslin Institute, R(D)SVS, University of Edinburgh, Easter Bush Campus, Edinburgh, Scotland, EH25 9RG, UK. [2]School of Biodiversity, One Health & Veterinary Medicine, College of Medical, Veterinary & Life Sciences, University of Glasgow, Glasgow, G12 8QQ, UK. [3]Department of Biosecurity Ecosystems and Veterinary Public Health, Makerere University, Kampala, 7062, Uganda. [4]Ministry of Agriculture and Animal Industry and Fisheries Uganda, Plot 16-18, Lugard Avenue, Entebbe, 102, Uganda. ✉e-mail: adrian.muwonge@roslin.ed.ac.uk

**Fig. 1 | Relationship between phenotypic and genotypic resistance in two pig production systems of Uganda, semi-intensive and free-range.**
**a** Proportions of *Escherichia coli* and *Klebsiella* spp. isolates along an antibiotic susceptibility scale, whereby "Zero" indicates pan-susceptible and "Seven" pan-resistant (to seven antibiotics).
**b** Comparison between this antibiotic susceptibility scale and the normalised copy number of genes *tetB* and *tetQ* (tetracyclines), *ermB* (tylosin) and *dfrA1* (trimethoprim). R is the correlation coefficient. The number on the bars represents the number of isolates per group.

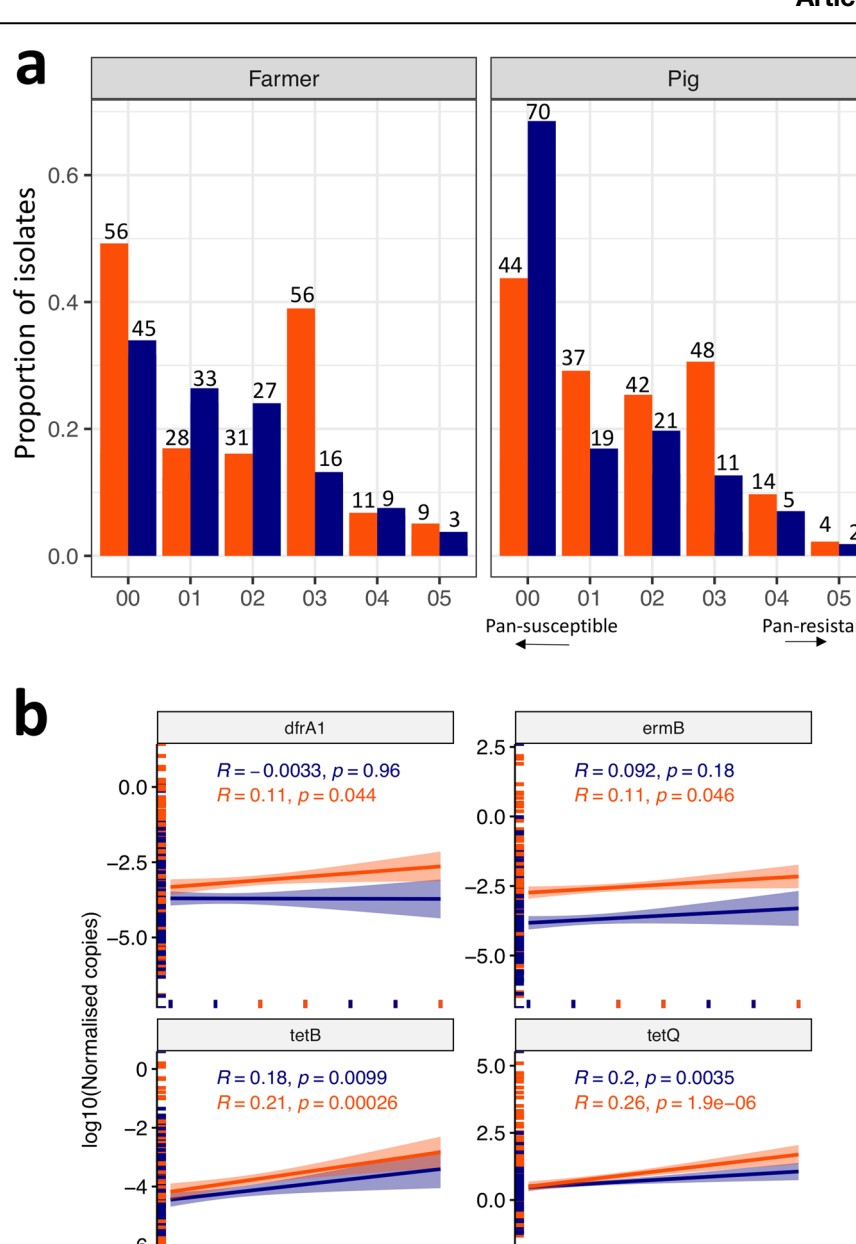

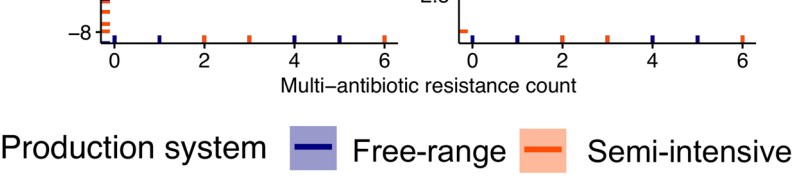

## Results

### Characteristics of AMR phenotypes and gene carriage in gut of farmers and pigs

We sampled 35 farmer-pig pairs in semi-intensive and free-range pig production systems, respectively (Supplementary Table S1). We observed a strong relationship between counts of multi-antibiotic resistance phenotypes to commonly used antibiotics (tetracyclines, trimethoprim/sulfamethoxazole, streptomycin, nalidixic acid, gentamicin and ciprofloxacin) and the selected gene *tetB* and *tetQ* (tetracyclines), *ermB* (tylosin) and *dfrA1* (trimethoprim) copy count in the microbiota from which bacteria were recovered (Fig. 1). The strength of this relationship (slope) varies depending on the AMR gene (Supplementary Fig. S1), production system and host (Fig. 1, Supplementary Table S1). In this case, the slope in semi-intensive systems was steeper, being 2 and 1.3 times higher than in free-range systems

for *tetQ* and *ermB*, respectively. Additionally, we observed a distinct difference in *ermB* carriage levels between the two production systems.

### Differences in AMR dynamics at sentinel bacterium-level

We recovered 668 sentinel bacteria of which *E. coli* comprised 64% and *Klebsiella* spp 36% of the samples cultured (Supplementary Fig. S1, Table S1). In our statistical models investigating drivers of resistance (Table 1), *Klebsiella* spp. exhibited significantly lower resistance (OR = 0.52, 95%CI: 0.42–0.64, $P < 0.001$) than *E. coli*, regardless of host and production system (Table 1, model 4; Table 2; Fig. 2a, Supplementary Fig. S2). We estimated a 20% prevalence (95% CI: 14.9–27.5) in farmers which was nearly twice that observed in free-range pigs (Fig. 2c). Compared to farmers, swine-exposed references (that we sampled to represent differing levels of pig exposure, from frequent to no contact, compared to farmers) were less

**Table 1 | Statistical models used for the analyses related to each of the objectives of this study, including the type of model and variables and interactions tested**

| Objective | Type of model | Main (fixed) effects | Interactions (fixed) effects[a] | Random effects |
|---|---|---|---|---|
| Model 1 A - Factors associated with phenotypic antibacterial resistance<br>Outcome variable - An isolate expressing resistance or not<br>Model 1B- Factors associated with phenotypic antibacterial resistance<br>Outcome variable - An isolate expressing resistance to a specific antibiotic (e.g., trimethoprim-sulfamethoxazole and streptomycin)<br>Model 1B is also used to analyse AMR for sentinel bacteria individually | Binomial GLMM | Model 1 A - Bacterium species (*Escherichia coli* /*Klebsiella*), production system (semi-intensive/free-range), host species (farmer/pig)<br>Model 1B - Bacterium species (*Escherichia coli* /*Klebsiella*), production system (semi-intensive/free-range), host species (farmer/pig), visit[c], pig breed, cleaning, AMR gene and treatment | Model 1 A - Host species x production system, production system x bacterium species, host species x bacterium species, production system x host species x bacterium species<br>Model 1B - Null | Model 1 A - Farm ID[b], visit[b], antibiotic type (tetracycline/ ciprofloxacin/ gentamycin/ Chloramphenicol/ streptomycin/ nalidixic acid/ trimethoprim-sulfamethoxazole)<br>Model 1B - Subcounty & visit[Δ] |
| Model 2 - Factors associated with antibacterial resistance gene.<br>Outcome variable is normalised copy number of genes *ermB* (tylosin), *tetB* and *tetQ* (tetracyclines) and *dfrA1* trimethoprim/sulfamethoxazole)<br>Model 3 - Factors associated with antibacterial resistance gene.<br>Outcome variable is normalised copy number of farmer at time T | Gaussian GLMM<br>Gaussian GLMM | Gene (*ermB*/*tetB*/*tetQ*/*dfrA1*), production system (semi-intensive/free-range), host species (farmer/pig)<br>Antibacterial resistance gene carriage - Outcome is normalised copy number of pig at time T + 1, gene (*ermB*/*tetB*/*tetQ*/*dfrA1*), production system (semi-intensive/free-range | Host species x production system<br>Host species x gene<br>Production system x gene<br>Host species x production system x gene<br>Host species x production system x gene | Farm ID[b], visit[b] |
| Model 4 - Farmer- and pig-associated factors in productions systems explaining the observed phenotypic resistance<br>Outcome variable- an isolate expressing resistance or not | Binomial GLMM | Production (semi-intensive/free-range), host species (farmer/pig),<br>Marital status (single/married/widow/divorced), bacterium species (*E. coli*/*Klebsiella*), pig breed (exotic/local/mixed breed), antibiotic use in farmers or pigs (yes/no), pig housing (indoor/outdoor), cleaning frequency (never/daily/bi-weekly/ weekly), visit#, Log10 (copy number of *tetQ*/*tetB*/*ermB*/ *dfrA1*) | Host species x production system | Farm ID[b], visit[b] |
| Model 5 - Transmission<br>Outcome variable -transmission event yes or no, i.e., a farmer and pig sharing a multi-drug resistance (MDR) pattern | Binomial GLMM | Log10 ratio (copy number of *tetQ*/*tetB*/*ermB*/ *dfrA1*) carried by farmers and pigs, time lag between a farmer and pig sample,<br>if farmer and pig were in the same household, and within and between production systems | NA | |

The variables used in models 1–4 are also used in the conditional inferencing framework to examine the structural relationships and importance of each variable.

*GLMM* Generalised Linear Mixed Model.

[a]In order to allow for the possibility that differences among bacterial species in levels of phenotypic AMR resistance might be host- and production system-specific, all two-way and three-way interactions were fitted in the initial model prior to selection. Similar reasoning motivated fitting all two- and three-way interactions in the gene carriage model.

[b]In order to account for non-independence between repeat observations from the same household, and to allow for variation between time points, random effects of Farm ID and sampling time point were fitted in most models. Modelling time as a random effect allows responses to vary between time points in a non-trend like way.

[c]This is a time continuous variable; Δ is a categorical variable in the random part of the model.

likely to carry resistant sentinel bacteria (OR = 0.41, 95% CI: 0.21–0.81, P = 0.01) (Supplementary Table S3). Compared to ciprofloxacin, trimethoprim/sulfamethoxazole (OR = 13.75, 95% CI: 8.39–22.64, P < 0.001), tetracycline (OR = 12.2, 95% CI: 8.04–18.55, P < 0.001) and streptomycin (OR = 6.79, 95% CI: 4.48–10.40, P < 0.001) showed the highest resistance levels (Fig. 2a, b; Supplementary Tables S2, 3). Individual antibiotic analyses revealed distinct resistance drivers (Supplementary Tables S4, 5). For example, local pig breeds exhibited lower resistance to tetracycline and trimethoprim/sulfamethoxazole. Higher resistance to streptomycin (OR = 1.48, 95% CI: 1.00–2.18, P = 0.047) and tetracycline (OR = 1.97, 95% CI: 1.22–3.18, P = 0.006) was linked to high copy number of *tetQ*, while elevated *tetB* levels were associated with resistance to trimethoprim/sulfamethoxazole (OR = 1.23, 95%CI: 1.02–1.47, P = 0.027) and chloramphenicol (OR = 1.33, 95% CI: 1.04–1.71, P = 0.025).

### Table 2 | Results of a mixed-effect logistic regression model (Model 1, Table 1) of phenotypic resistance of *Escherichia coli* and *Klebsiella* species isolated from pigs and farmers in free-range and semi-intensive production systems of Uganda

| Variables | Levels | Odds ratio (95% CI) | P Value | Variance |
|---|---|---|---|---|
| **Fixed effects** | | | | |
| Production system | Free-range | 1 (ref) | | |
| | Semi-intensive | 1.31 (0.93–1.84) | 0.120 | |
| Bacteria | *E. coli* | 1 (ref) | | |
| | *Klebsiella* | 0.41(0.33–0.51) | <0.001 | |
| Host | Farmer | 1 (ref) | | |
| | Pig | 0.84 (0.66–1.06) | 0.147 | |
| Interaction | Peri-urban: Pig | 0.56 (0.37–0.86) | 0.007 | |
| **Random effects** | | | | |
| Household ID | | | | 0.15 |
| Visits | | | | 0.13 |
| Antibiotic | | | | 1.02 |

*CI* Confidence Interval.

### Differences in AMR dynamics at host level

Most farmers were males (68.0%) and had attained primary and secondary education (82.8%) (Supplementary Table S1). Overall, 52.3% of the pigs were housed, fed on a mixture of feeds (74.0%) and swill (13.0%). Exposure to antibiotics a fortnight before each visit was comparable between farmers and pigs (Supplementary Table S1), but exposure was higher in pigs in semi-intensive farms (62.3%).

AMR was lower in free-range (OR = 0.56, 95% CI: 0.37–0.86, P = 0.007) compared to semi-intensive systems (Table 2, Fig. 2a). Similarly, pan-susceptible bacteria were more common in free-range compared to semi-intensive systems (Fig. 1a, Supplementary Fig. S3). Levels of AMR in farmer-pig pairs sampled in semi-intensive systems were similar, while there were significant differences in free-range systems (Fig. 2c), where pigs were less likely (OR = 0.44, 95% CI: 0.28–0.70, P < 0.001) to be colonised by resistant bacteria compared to farmers (Fig. 2a, c). We observed a higher carriage of the *ermB* gene, which increased with multi-antimicrobial resistance (MDR) (Fig. 1b, Supplementary Figs. S1, 3) - we define MDR as resistance to at least three antibiotic classes. In general, the copy number of genes carried in the gut of farmers from semi-intensive production systems was higher than in their pigs. Here, the largest difference was observed for *tetB*, while for the swine reference group it was for *dfrA1* (Supplementary Fig. S1). There was a temporal dimension in levels of AMR (OR = 1.15, 95% CI: 1.02–1.31, P = 0.024) (Figs. 2e, 3b), with an average increase of 0.76% in

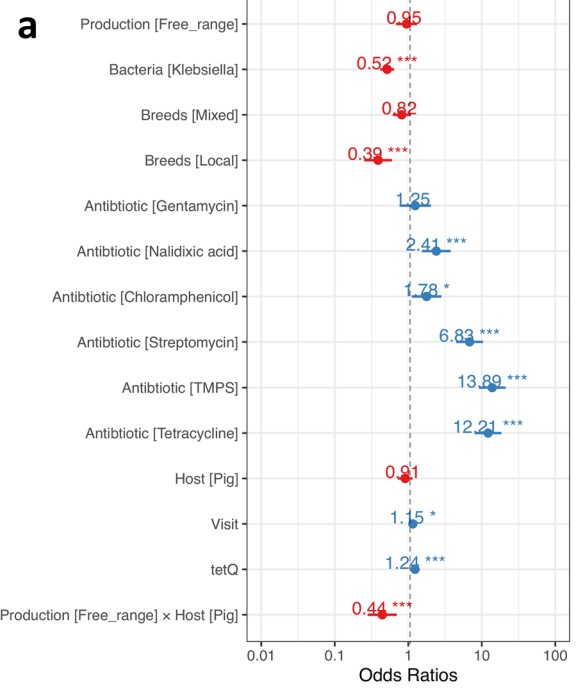

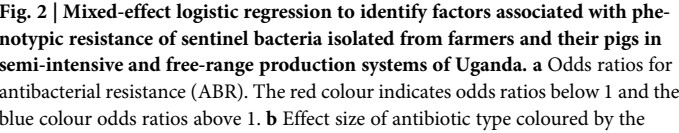

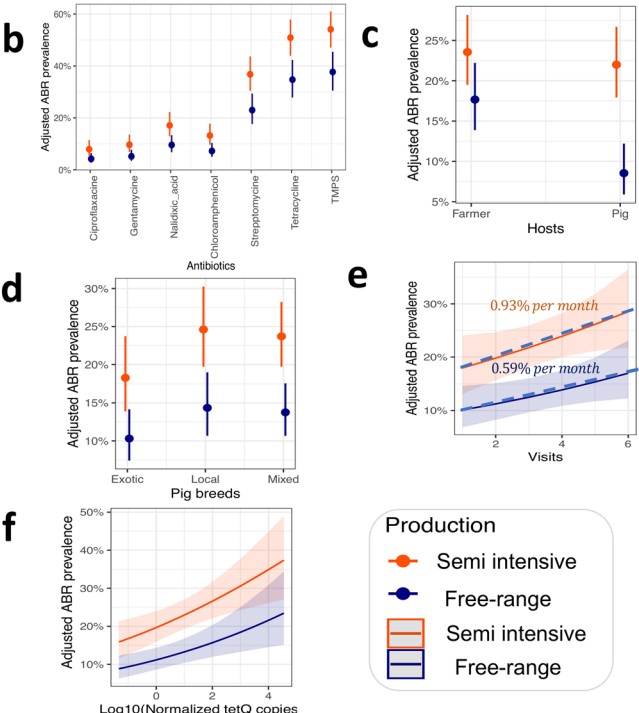

**Fig. 2 | Mixed-effect logistic regression to identify factors associated with phenotypic resistance of sentinel bacteria isolated from farmers and their pigs in semi-intensive and free-range production systems of Uganda. a** Odds ratios for antibacterial resistance (ABR). The red colour indicates odds ratios below 1 and the blue colour odds ratios above 1. **b** Effect size of antibiotic type coloured by the production system. **c** Effect size of the host. **d**–**f** Effect size of pig breed, visit as temporal variable, and *tetQ* (tetracyclines) normalised copies. We estimated that, on average, ABR is increasing at 0.76% per year, with an higher (0.93%) increase in semi-intensive systems. TMPS Trimethoprim/sulfamethoxazole.

the twelve months of the study (Fig. 2e and Supplementary Fig. S4). The production system also influenced the strength of the relationship between the number of antibiotics to which bacterial isolates were resistant and the copy number of genes in the gut microbiota from which such isolates were recovered (Fig. 2f, Supplementary Fig. S5). Furthermore, the conditional inference tree (CIT) analysis suggests that cleaning frequency was the primary predictor of copy number for all AMR genes except *tetB*, which was mainly influenced by the production system [Supplementary Figs. S6, 7, 8]. Overall, previous medication, breed, and production system were also important predictors of AMR gene carriage.

Similarly, copy numbers of *tetQ* and *tetB* increased across the six visits, with significant differences between semi-intensive and free-range farms (Figs. 2, 3b). In general, a tenfold increase in *tetQ* copy number was associated with 1.24 odds of a colonising bacterium being resistant (Fig. 2f; Supplementary Tables S2, 6, 7, 8).

Regarding pig-related factors associated with AMR, local breeds were less likely to carry resistant bacteria (OR = 0.39, 95% CI: 0.25–0.60, $P > 0.001$) compared to exotic breeds. Pigs on semi-intensive farms were 2.2 time more likely to carry resistant sentinel bacteria compared to those in free-range systems (Fig. 2a, c).

While our regression modelling approaches revealed no association between AMR and previous medication in farmers or their pigs, the CIT approach shows that prior medication was a significant predictor of AMR, particularly to tetracycline, streptomycin and sulfa-trimethoprim [Supplementary Figs. S5, 6]. Critically, the former approach revealed that most of the unexplained variation was at farm level (Supplementary Tables S2, 3, 7, 8).

## Potential for transmission events at the human-pig interface

We investigated whether gene carriage in farmers influences that of their pigs and vice versa and found no significant correlation between gene carriage by a farmer at visit T and their pig at T + 1 (model 5, Table 1 and Supplementary Tables S6, 8). To further explore the potential for cross-species transmission, we examined the sharing of multi-drug resistant profiles. We observed sixteen unique MDR profiles both in semi-intensive and free-range productions systems, although the greatest number was observed in semi-intensive systems (Supplementary Fig. S9). We compared levels of MDR sharing (potential transmission events, PTEs) at farm level, and within and between production systems (model 4, Table 1, Supplementary Fig. S9). Overall, we detected 25.1% (95% CI: 23.1%–27.1% of PTEs (Supplementary Table S9, Fig. S10), with the greatest number (38.2%) occurring at farm level followed by within (30.7%) and between (11.1%) production systems (Fig. 4a, Supplementary Fig. S8). Therefore, the likelihood of a PTEs was higher on farm (OR = 3.16, 95% CI: 1.52–5.40, $P = 0.004$) and within (OR = 3.13, 95% CI: 2.1–4.3, $P < 0.001$) than between production systems, respectively (Fig. 4b and Supplementary Figs. S9, 10). The likelihood of PTEs was higher (OR = 4.34, 95% CI: 2.85–6.01, $P < 0.001$,)) if MDR sharing between a pig and a farmer on a given farm was detected during the same visit (time lag 0) than during different visits (Fig. 4b). Furthermore, there was a significant relationship between the copy number of genes carried by a farmer and their pig and the probability of PTEs (Fig. 4c). This probability was higher if a farmer carried a greater copy number of genes *dfrA1* and *tetQ* than their pig, while the reverse was true when the carriage of *ermB* was higher in their pigs.

## Discussion

To investigate the drivers of AMR, particularly in the context of a potential shift from free-range to semi-intensive livestock production in LMICs, we tracked AMR to seven antibiotics in two bacterial sentinels as well as gene carriage in the guts of farmers and their pigs over one year in peri-urban and rural Uganda. The former setting represents higher antibiotic use[19] and intensification compared to the latter. As such, the two settings serve as proxies for semi-intensive and free-range systems, respectively. We observed significant differences in AMR at the level of sentinel bacteria and hosts, and at temporal and spatial scales. The spatial scale differences were

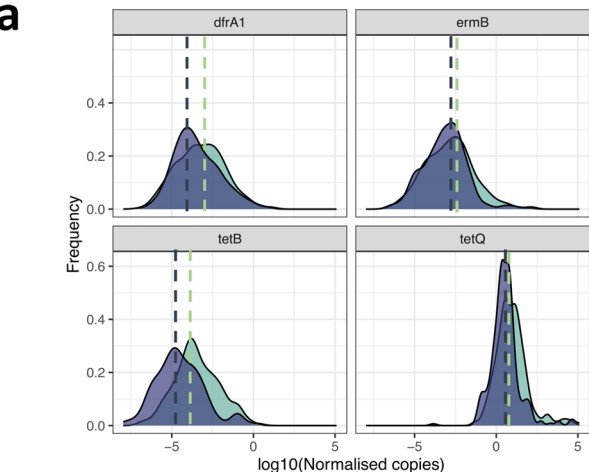

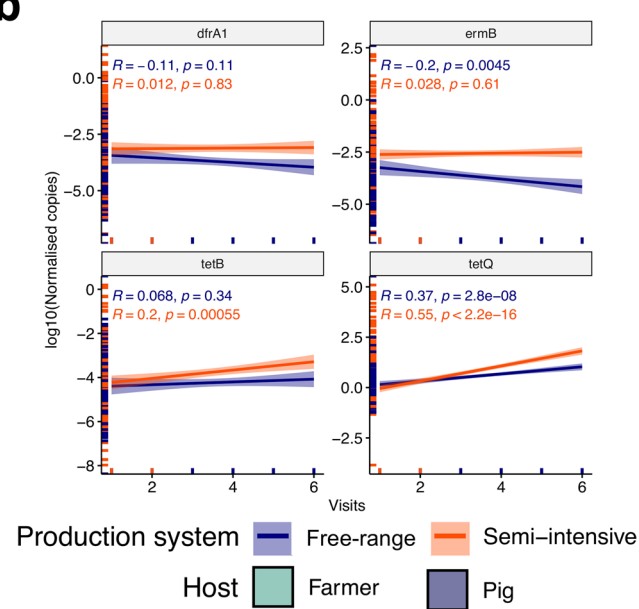

**Fig. 3 | Temporal dynamics of antibacterial-resistance genes in pig production systems of Uganda. a** Is the distribution of the normalised copy number of antibacterial resistance genes *tetB* and *tetQ* (tetracyclines), *ermB* (tylosin) and *dfrA1* (trimethoprim) and variation in counts across study farmer and pig populations. Most individuals fall at the peak of the curve (at the dotted line). **b** Relationship between normalised copy number of a gene and visit number indicating the temporal scale. A visit was repeated every two months for a period of one year with a total of six visits undertaken. R is the correlation coefficient.

likely associated with production system-level drivers. However, since only two districts were studied, the interactions between production systems and geography were not adequately delineated. At the temporal scale, we observed a steady increase in resistance levels over the period of the study. Furthermore, the potential for transmission events was higher at farm level and in semi-intensive production systems, warranting further investigation using granular, genomic-based transmission analysis. Our overall key findings are that AMR, gene carriage and the potential for transmission events are higher in peri-urban/semi-intensive production systems, indicating increased AMR risks associated with the transition from free-range to intensive farming and rural-urban migration.

## Mono- and multi-antibiotic resistance is associated with higher AMR gene carriage

Our design uniquely enabled us to explore AMR at multiple levels. Firstly, the longitudinal sampling we employed allowed us to characterise temporal

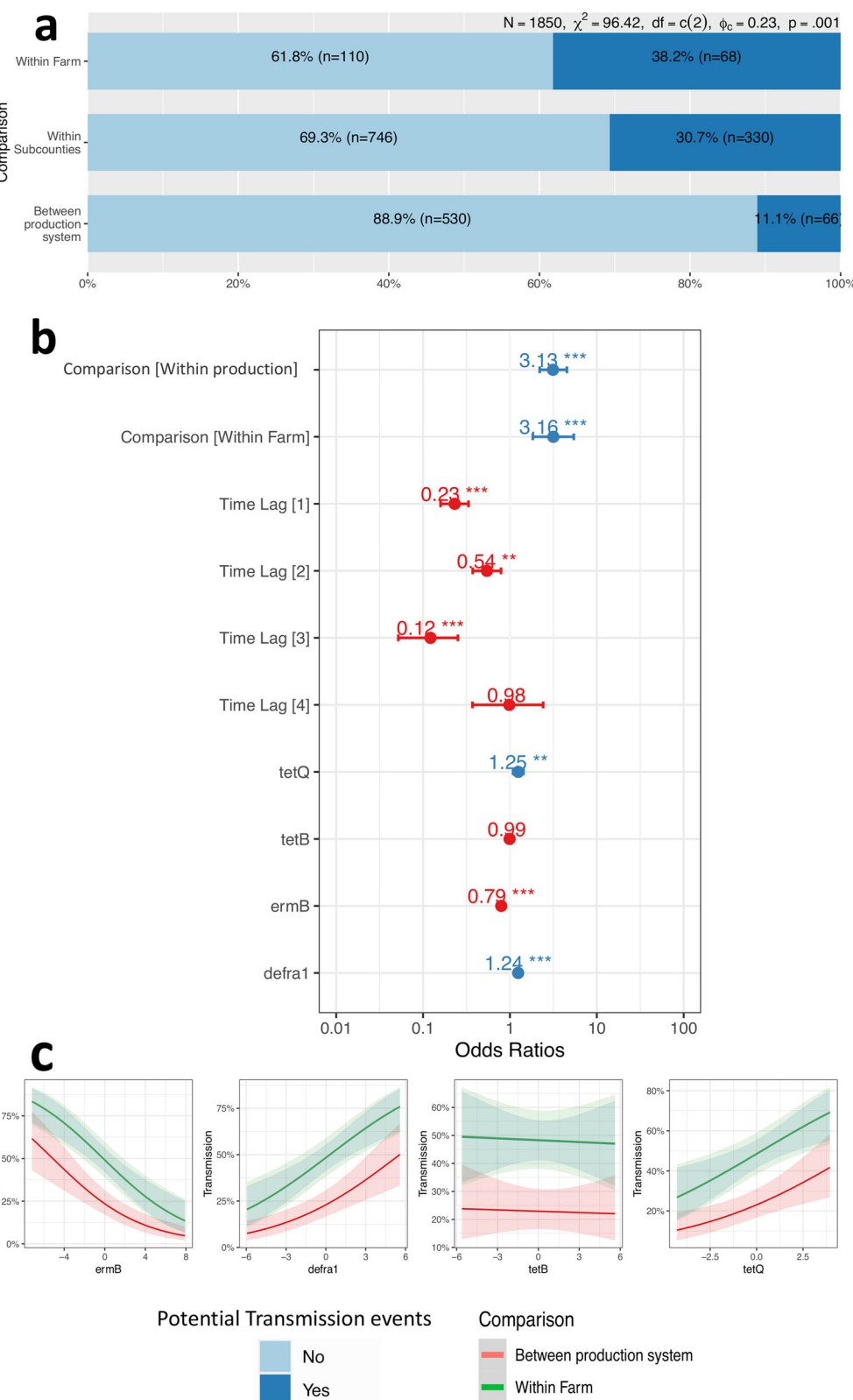

**Fig. 4 | Drivers of potential transmission events (TEs) determined using a generalised logistic regression model (Model 5, Table 1). a** Proportion of TEs at farm level, within sub-county and between production systems. **b** Model output. The time lag refers to the number of visits between compared multi-drug-resistant profiles linking a farmer to a pig. For example, time lag 0 refers to shared profiles within the same visit. The gene variable measures the log10 ratio carriage by a farmer compared to a pig, i.e., a log10 ratio >1 indicates that the farmer carries more copy numbers of a gene than their pig (**c**), while a log10 ratio equal to 1 means that the carriage of a pig and a farmer is the same. The antibacterial resistance genes examined were *tetB* and *tetQ* (tetracyclines), *ermB* (tylosin) and *dfrA1* (trimethoprim).

changes in AMR, highlighting a 0.76% increase over the one-year study period. This estimate is slightly higher than the national estimate of 0.50%, derived from a four-year analysis of AMR in clinical pathogens that also indicated a growing trend[20]. Secondly, we were able to compare the phenotypic resistance of sentinel bacteria with the copy number AMR gene carriage of farmers' and pigs' resident gut microbiome. We found a positive correlation, i.e., the number of antibiotics to which a sentinel bacterium is resistant increases with the copy number of selected AMR genes. The strength of this relationship varies depending on the type of gene and production system. Overall, this supports our assumption of a selection pressure-mediated phenotype. Elsewhere, it has been shown that the copy number of genes such as $bla_{KPC-2}$ is associated with the proliferation of *Klebsiella pneumoniae* carbapenemase (KPC)[21] and, therefore, the AMR clinical phenotype. In contrast, the copy number of fungal resistance genes has been reported to play a limited role in the variability of their AMR phenotypes[22]. Interestingly, a similar relationship between MDR and AMR gene carriage has been observed in ornamental fish, with the highest levels of MDR found in tropical fish species[23] highlighting the fundamental nature of this association. One can therefore argue that the observed MDR phenotype represents and reflects selection pressure driven by AMU. Other studies have shown a reduction in such AMR phenotypes when the underlying AMU-mediated selection pressure is removed. We also observe a right-skewed MDR prevalence distribution, suggesting a fitness cost of MDR phenotype to bacteria[24] particularly *Klebsiella* spp, that warrants further investigation.

**Differential occurrence of tetracycline selective mechanisms of action.** In this study, *tetQ* had the highest copy number, nearly six-fold higher than *tetB*, suggesting a stronger selection for ribosomal protection over tetracycline efflux[25]. This difference is also reflected in relation to potential transmission events (PTEs) as discussed below. Overall, this indicates high use of this antibiotic, although the distinct carriage level of *tetB* between farmers and their pigs my reflect the difference in lifetime (humans live longer than pigs) and/or legacy use. Further investigation on the role of gene copy number in the AMR phenotype variability, including multi-drug resistance, is important to identify novel control strategies. By contrast, rural residents of India are reported to carry *Enterobacteriaceae* species exhibiting high levels of MDR[26] which suggest geographic-specific epidemiology of AMR/MDR. In this study, we compare the levels of two *Enterobacteriaceae* species and note that *E. coli* exhibits higher AMR levels than *Klebsiella* species. This pattern was consistent among farmers and their pigs, regardless of the production system, and is probably linked to *E. coli*'s superior capability to acquire resistance via horizontal gene transfer[27].

**Peri-urban/semi-intensive farms exhibit disproportionately higher AMR levels**

Distinct AMR distribution patterns between production systems are shaped by spatial dynamics, host characteristics, and differences in pathogen selection pressures[28]. One such host characteristic is the rapid human population expansion. This expansion not only alters the transmission contact structure between species but also the demand for animal protein[29,30] which must be produced on a non-elastic land resource. Farming communities respond to such demand by producing more per unit land area (intensification or semi-intensification) and closer to the market, i.e., peri-urban areas. Elsewhere, intensification is associated with higher antimicrobial use to boost production[31]. Indeed, in this study, we observe higher levels of AMR in the peri-urban/semi-intensive system, which suggest a strong link between both location and livestock production context. Our conditional inference tree analysis reveals that in semi-intensive systems, the absence of prior pig medication was linked to lower resistance, particularly among local pig breeds. Conversely, when pig medication is reported, it correlates with higher resistance to tetracycline. However, studies across sub-Saharan Africa suggest that this relationship is more complex than just AMU alone. The production system differences are likely much more

complex than just AMU, consistent with other studies in agricultural communities of sub-Saharan Africa. For example, in Tanzania, antibiotic use was not directly associated with AMR but to the quality of community water sources, consumption of contaminated milk, and livestock production systems[32]. A similar observation on water quality and AMR has been reported in Uganda[33]. It is noteworthy that, although we qualitatively surveyed farmers on their antibiotic use, we had no data on quantities used. This limited further investigation of a possible association between the AMR patterns observed and antibiotic use.

**Farmers exhibit higher AMR levels than their pigs**

When compared to swine exposure references, bacteria retrieved from farmers and their pigs exhibited higher levels of antibiotic resistance across all seven antibiotic classes, especially those isolated from farmers. This difference between pigs and farmers was greatest in the rural/free-range pig production system. This result may suggest differential levels of transmission risk between farmers and pigs in the production systems examined with the risk being lower in free-range systems, as we discuss further below. This may reflect differences in antibiotic use between humans and livestock[31,34]. Globally, antibiotic use in livestock is declining[34]. Levels of use in Africa have consistently been lower than in Europe and Asia where use in livestock has historically exceeded that in humans[34,35]. The opposite may be true in Africa, as reflected in this study by higher levels of AMR in farmers regardless of production system. Previous studies in Southwest Uganda have also shown higher AMR levels in humans compared to cattle[33].

**The potential for transmission is higher on semi-intensive farms**

The observed characteristics of the AMR phenotype, i.e., monoresistance and MDR, between production systems may arise from a) selection pressure due to antibiotic use and/or (b) distinct transmission risks inherent to the system at farm level. We investigated this issue in our study and found that the overall sharing of MDR patterns between farmers and their pigs—our proxy for a potential transmission event—was estimated at 9.8%. Indeed, we found that potential transmission events (PTEs) were more likely at the farm level than at the "between production system" spatial scale, especially when sentinel bacteria were recovered within the same temporal window (during a specific farm visit). This aligns with the epidemiological principle that temporal and spatial proximities are key factors in transmission. These PTEs were common on farms where farmers carry significantly more trimethoprim-resistance-conferring *dfrA1* genes[25] than pigs. A similar pattern was observed on farms where pigs carried higher levels of *ermB* which confers resistance to macrolides such as tylosin by restricting ribosomal binding sites, thereby blocking protein synthesis[36]. This suggests differential use of trimethoprim/ sulfamethoxazole and tylosin by farmers and their pigs, respectively. Indeed, tylosin is an antibiotic exclusively used in veterinary medicine and the most used in pig farming globally[37]. We recognise the limitations of our coarse transmission unit (MDR) which likely introduce uncertainties. Thus, we refer to these as PTEs and recommend that more precise genomic-based tools are used for future analysis. Interestingly, the gene carriage of a farmer at a given time point was not correlated with that of their pig at the subsequent time point. This could be due to limited statistical power to detect a causal relation at that level within a two-month window. Despite these limitations, our focused longitudinal methodology provided increased depth per farm, subcounty and production system to reveal consistent AMR dynamics. However, the effects of production system and geographical differences remain unresolved. This highlights that additional factors not captured by our study design need investigating. Future longitudinal studies should integrate a broader range of variables across hosts, pathogens, and production systems in diverse geographical contexts.

In conclusion, our study highlights a disproportionately higher level of AMR and AMR gene carriage in peri-urban, semi-intensive pig production systems. Farmers in these systems carry more resistant sentinel bacteria compared to those in rural, free-range systems. Therefore, our findings suggest a potentially increased AMR risk in urban swine production in

Uganda, which may escalate with intensification as countries like Uganda respond to growing demand for animal protein. Our models show that farm-level factors drive most AMR variation, highlighting the need for targeted interventions at this level.

## Methods

### Ethics statement

Ethical approval was obtained from the: (1) School of Biomedical Sciences, Makerere University College of Health Sciences (REF-SBS-140); (2) Uganda National Council of Science and Technology (UNCST REF: HS103ES); and (3) Easter Bush Veterinary and Human Ethical Review Committees, University of Edinburgh (VERC-82-17 & HERC-133-17). Informed consent was obtained in writing from each participant. Therefore, this study was conducted in compliance with all relevant ethical regulations.

### Longitudinal field study

Design. We conducted a longitudinal cohort study encompassing pairs comprising a farmer and their sow pig on selected study farms. Pairs were followed bimonthly for a year (six time points) between November 2018 and October 2019 in two districts representative of semi-intensive (Kampala) and free-range (Mubende) production systems (Supplementary Fig. S11). Sows were used as they remain on farms for reproduction and long-term management. The pig production categorisation was based on factors such as stocking density, access to land resources, disposable income, access to improved breeding inputs, supplementation, market access and productivity (Supplementary Fig. S12).

Sample-size estimation. Thirty-five farmer-pig pairs were monitored in each production system. To estimate this sample size, we assumed an AMR prevalence risk ratio of 5.5 with 80% power at the 5% significance level, based on cross-sectional studies with 38% and 10% average prevalence in high- (semi-intensive) and low- (free-range) AMU systems, respectively. We accounted for a 10% attrition rate and clustering (design effect) and included both high-exposure (an abattoir worker handling pig gut evisceration) and low-exposure (individuals with no pig contact due to religious or cultural taboos) swine reference groups in each of the five sub-counties where the pairs resided (Supplementary Table S1). A total of 840 and 120 samples were expected from 70 pairs and 20 swine exposure references (10 swine and 10 non-swine exposure references) across six study visits.

### Assumptions and causal framework

We employed a causal framework (Fig. 5) to contextualise the emergence and transmission of AMR phenotypes in semi-intensive and free-range pig production systems. For instance, we assume that antibiotic use in pigs—shaped by factors such as breed, management practices, and seasonal variations—creates selection pressure in the gut, ultimately driving the development of AMR phenotypes. The same assumption applies to the farmers. To this effect, we collected a total of 877 faecal samples, including 382 and 386 from pigs and farmers, respectively, and a further 109 samples from swine exposure references (Supplementary Table S1). These samples were subjected to culture, antibiotic sensitivity testing, and gene quantification using qPCR.

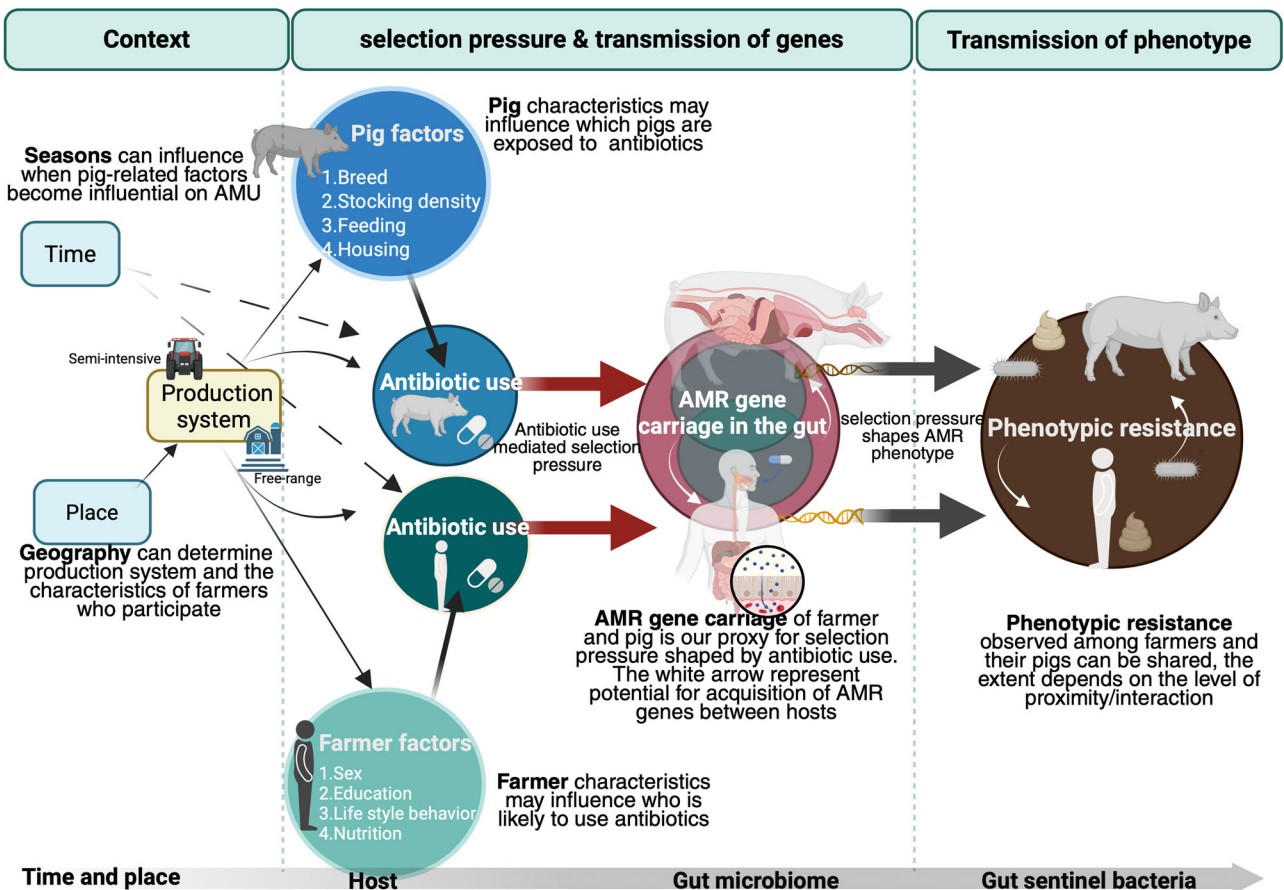

**Fig. 5 | Causal framework showing the potential drivers of antibacterial resistance (ABR) examined at the pig-human interface in Uganda.** Factors examined included antibiotic use, farmer- and livestock-associated factors, time of sampling and production systems, i.e., semi-intensive and free-range. Here, we focus on the context, drivers of selection pressure and the state of antibacterial resistance of sentinel bacteria. AMR antimicrobial resistance, AMU antimicrobial use.

## Microbiological analyses

Faecal samples, collected using a stool collection kit, were submitted to the Molecular Biology Laboratory at the Schools of Biomedical Sciences of Makerere University (Kampala, Uganda) for culture, sensitivity testing and DNA extraction. An aliquot was sent to the Makerere Central Veterinary laboratory for bacterial culture. DNA was sent to the Digital One Health Laboratory, Roslin Institute, University of Edinburgh for down-stream qPCR analysis.

**Phenotypic antimicrobial resistance.** Isolation of *E. coli* and *Klebsiella* spp. followed standard methodology. In brief, selective enrichment culturing using lauryl tryptose broth was followed by incubation at 37 °C for 18–24 h and streaking on MacConkey agar for further selective and differential growth, with identification based on morphology and standard biochemical test[38]. A single colony was subsequently selected for disc diffusion antibiotic susceptibility testing focusing on seven antibiotics: tetracycline, ciprofloxacin, chloramphenicol gentamycin, streptomycin, nalidixic acid and trimethoprim/sulfamethoxazole. The EUCAST (European Committee on Antimicrobial Susceptibility Testing) guidelines were used for interpretation[39]. These were selected based on their use in human and animal health in Uganda. We assumed that: (1) the phenotypic resistance of the selected colony represented the dominant sentinel population[40] and (2) if a phenotypic signal was detectable in the sentinel bacteria, which account for ~0.02% of gut flora, then this phenotypic resistance was likely universal to the microbiome or at least specific to the family *Enterobacteriaceae*[41]. We then extracted high molecular grade DNA, an aliquot of which was sent to The Roslin Institute for downstream qPCR analysis of gene carriage.

**Quantification of AMR gene carriage.** To compare the phenotype to genotype, we tracked copy numbers of AMR genes using qPCR relating them to the phenotype above. We targeted four genes which encode resistance to tylosin (*ermB*), tetracyclines (*tetB* and *tetQ*) and trimethoprim/sulfamethoxazole (*dfrA1*). We carried out 16s RNA gene quantification using standard qPCR methods[42] for gene taxa normalisation.

## Statistical analyses

Statistical modelling was conducted in R version 3.5.1 (https://cran.r-project.org) using generalised linear mixed models (GLMMs) using the binomial link with *lme4* package (v1.1-26). Our unit of analysis was the farm, while the outcome variables were the AMR of bacteria and AMR gene carriage. We tested for associations between outcome and explanatory variables as shown in Table 1, including their interactions, using likelihood ratio tests (LRTs), with the drop1 function in *lme4 package*, starting from the most complex model using the backward stepwise selection approach. The random effects were modelled as intercepts with Farm_IDs nested within a visit. The independence assumption was ensured by using a mixed- effect model using time and place as random variables, and independence by comparing the correlation of group means to the effects. Only significant variables ($P < 0.05$) were retained in the final model. Models, variables and interactions tested are summarised in Table 1.

In brief, we first compared the prevalence of AMR phenotypic resistance (proportion of isolates resistant to each selected antibiotic) across bacterial sentinel, host and production system (Model 1, Table 1). It is plausible that the variation in AMR between farmers and pigs is inherent to a production system affecting host gene carriage (Fig. 5). To investigate this, we used a binomial logistic regression model with AMR as the outcome variable. A backward model selection approach was adopted using the *drop1* function to remove variables with limited statistical relevance.

To determine factors associated with changes in gene copy counts (Model 2, Table 1), we used qPCR data of copy numbers of the four selected genes (*tetB*, *tetQ*, *ermB* and *dfrA1*) from the total DNA extracted from faecal samples from a farmer and their pig. These microbiome-wide genotypic characteristics were then compared to the phenotypic resistance expressed by the sentinel bacteria recovered from a sample. We also qualitatively examined the trend of gene copy count over time (Model 3, Table 1). To examine associations, the outcome variable was the normalised gene count (~copy number per bacteria). Normalisation was based on 16 s RNA copy numbers (copy number *tetB* of a sample/16sRNA copy number of a sample) with the resultant measure being an approximate number of gene copy per bacterium. Here, the assumption is that a bacterium carries a single gene copy[42]. The independence assumption was assessed by plotting the residues and fitted values. Normality was assessed using the quantile-quantile normality plots and independence by comparing the correlation of group means to the effects.

To investigate human- and livestock-associated factors driving the observed phenotypic and genotypic resistance in the two pig production systems, we expanded the model described above beyond location and bacterium and host type to include farmer gender, wealth and lifestyle indicators, and medication use, as well as pig-related factors (Model 4, Table 1).

To investigate if gene carriage in farmers interacts with that of their pigs and vice versa we modelled the gene carriage of a farmer at time (T) to that of their linked pig at time (T + 1) (Model 3, Table 1). Finally, we investigated signatures of potential transmission between farmers and pigs. We assumed that sharing a multi-drug-resistant strain between a pair in a household is a rare event, but if it is to happen, it is most likely at farm level, and therefore, when detected, represents a potential transmission event. We compared prevalence of potential transmission events at household, subcounty and district levels. With a binary outcome of presence and absence of a potential transmission event, we investigated the factors associated with this outcome variable (Model 5, Table 1). We used the conditional reference trees (CIT in *partykit* package in R) as non-parametric decision tree based on conditional inference principles to assess the structure and importance of variables in models 1-4.

## Reporting summary

Further information on research design is available in the Nature Portfolio Reporting Summary linked to this article.

## Data availability

The datasets used for analysis are accessible at https://datashare.ed.ac.uk/handle/10283/8847.

## Code availability

The R-code used for analysis are accessible at https://doi.org/10.7488/ds/7792.

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

## Acknowledgements

We acknowledge the invaluable support provided by the Mubende district, particularly the district health services and the veterinary production departments. Furthermore, we are indebted to all the farmers who participated in this study. We extend our gratitude to Dr Jolinda Pollock for her contribution to the qPCR analysis as a core scientist at the Roslin Institute.

## Author contributions

The study was conceptualised by A.M. and B.M.C.B. Data were curated by A.M. and B.M.C.B. and funding for the BBSRC Future Leader Fellowship and T.K.'s Msc was acquired by A.M. and T.L., respectively. Field methodology was developed by A.M., and the project implementation and administration were responsibility of L.K. and M.K., supervised by C.K. Data analysis and validation were performed by T.K., P.C.D.J., A.M., and T.L. Data visualisation was done by A.M. and original draft writing, reviewing, and editing involved all authors.

## Competing interests
The authors declare no competing interests.
