## [Transparent Peer Review file · Communications Earth & Environment]

Pig production system drivers of antibiotic resistance in Uganda

Corresponding Author: Dr Adrian Muwonge

Version 0:

Decision Letter:

Dear Dr Muwonge,

Your manuscript titled "Pig production system drivers of antibiotic resistance in Uganda" has now been seen by 3 reviewers, whose comments are appended below. You will see that they find your work of some potential interest. However, they have raised quite substantial concerns that must be addressed. In light of these comments, we cannot accept the manuscript for publication, but would be interested in considering a revised version that fully addresses these serious concerns.

In revision, please address the following editorial thresholds:

* Provide a compelling finding about impact of shifts in pig production system on antibiotic resistance in Uganda, by a thorough analysis of potential drivers.

* Clarify the methods and describe them at a level of detail that will allow independent researchers to replicate your results and fully acknowledge the limitations of your approach so that results can be easily interpreted.

We hope you will find the reviewers' comments useful as you decide how to proceed. Should additional work allow you to address these criticisms, we would be happy to look at a substantially revised manuscript. If you choose to take up this option, please either highlight all changes in the manuscript text file, or provide a list of the changes to the manuscript with your responses to the reviewers.

When resubmitting, please provide a point-by-point response to the reviewers' comments. Please submit your responses as a separate file, distinct from your cover letter where you can add responses to the Editors' comments that you do not want to be made available to the reviewers. Word files are preferred. We recommend that any figures, tables or graphs that are included in the response to reviewers are also included in the main article or Supplementary Information.

If the revision process takes significantly longer than three months, we will be happy to reconsider your paper at a later date, as long as nothing similar has been accepted for publication at Communications Earth & Environment or published elsewhere in the meantime.

Please use the following link to submit your revised manuscript, point-by-point response to the reviewers' comments with a list of your changes to the manuscript text (which should be in a separate document to any cover letter), a tracked-changes version of the manuscript (as a PDF file) and any completed checklist:

Link Redacted

Please do not hesitate to contact us if you have any questions or would like to discuss the required revisions further. Thank you for the opportunity to review your work.

Best regards,

Mengjie Wang
Associate Editor
Communications Earth & Environment
@CommsEarth

EDITORIAL POLICIES AND FORMAT

If you decide to resubmit your paper, please ensure that your manuscript complies with our editorial policies and complete and upload the checklist below as a Related Manuscript file type with the revised article:

Editorial Policy Policy requirements
(Download the link to your computer as a PDF.)

- Behavioural and social science
- Ecological, evolutionary & environmental sciences
- Life sciences

<https://www.nature.com/documents/nr-reporting-summary.zip>

For your information, you can find some guidance regarding format requirements summarized on the following checklist: (<https://www.nature.com/documents/commsj-phys-style-formatting-checklist-article.pdf>) and formatting guide (<https://www.nature.com/documents/commsj-phys-style-formatting-guide-accept.pdf>).

REVIEWER COMMENTS:

Reviewer #1 (Remarks to the Author):

This paper aim to assess whether change in production system (from free-ranging pigs at small scale to pigs kept in semi-intensive system) drives antibiotic resistance in a low-income country. The Key findings are that there are higher resistance levels (phenotypic and genetic) in pigs and farmers at semi-intensive farms compared to pigs and farmers at the small-scale farms. This in turn may suggest that there is a risk of over-all increase in antibiotic resistance if the transition to more intensive pig production systems is not well managed.

This study is not unique -comparisons of AB-burden among farmers and pigs between different pig farming systems has been conducted in other low/middle income countries (e.g. Lunha et al. "Antimicrobial Resistance in Fecal Escherichia coli from Humans and Pigs at Farms at Different Levels of Intensification." *Antibiotics* (Basel, Switzerland) vol. 9,10 662. 30 Sep. 2020, doi:10.3390/antibiotics9100662. , Hallenberg et al. "Antibiotic use in pig farms at different levels of intensification-Farmers' practices in northeastern Thailand." *PLoS one* vol. 15,12 e0243099. 11 Dec. 2020, doi:10.1371/journal.pone.0243099; Hickman, et al. "Exploring the Antibiotic Resistance Burden in Livestock, Livestock Handlers and Their Non-Livestock Handling Contacts: A One Health Perspective." *Frontiers in microbiology* vol. 12 651461. 20 Apr. 2021, doi:10.3389/fmicb.2021.651461). However, despite this lack of novelty the paper provides relevant information about the situation in Uganda, stressing the importance to manage the intensification of pig farming systems in a sensible way.

In addition to these very general comments, this reviewer provides the following comments.

One conceptual flaw of the paper is that the authors use sharing of antibiotic profile (different aspects) in pigs and humans as a proxy for transmission. This might be, but there is not solid scientific evidence behind this. So please revise throughout the ms say "sharing" instead of "transmission". You may, of course, speculate that in the discussion that transmission occurs.

One small editorial remark, the superscript numbering for the author affiliations seems to be in wrong order.

Comments by line in the ms

L: 35-37, it is confusing to which percentage refers to what? Please clarify

L 46: There are substantial amounts of publications and strategies on control strategies for ABR in livestock – please revise.

L 48: Is a publication from 2018 really recent? There must be more recent publications supporting this very general statement.

L 51: Usually increased incomes are added to this list of factors – likely more important than general migration.

L56-57: Please note that increased intensification/productivity not necessarily have to be followed by increased ABU as prophylaxis (e.g. Denmark). Please nuance.

L 102: do the 23% refers to pigs and humans as well.

L 104: please provide reference group data here as well. Also, the reference groups seems to be very heterogeneous (L248-249)

L 104-L109: Over all, or from humans in semi-intensive, pigs in free-ranging etc? Please clarify also in the running text here.

L 112: "pig has used antibiotics" sounds odd, please rewrite

L 128: Same as L 104

L 145: the common definition of MDR is that the bacteria is resistant to antibiotics of 3 different classes – please change and check data.

L 166: it is obvious that production systems are associated with production systems, pls rewrite.

L172: "ABR risk", is very much jargon. Please be more elaborate.

L 173-74: , "likely due to.." statement is very vague/general and doesn't add to the discussion, please omit.

L 177-78: Highly speculative and general, not supported by the data. Pls omit.

L 182-184: This is contradictory and confusing. Pls rewrite.

L 185: "This finding", please specify.

L 187-L192: Please shorten this section -it is hard to follow the reasoning.

L 198: is 2017 recent? Please look for more recent reviews.

L 203: Reference "27" is an odd one?! Please delete.

L 213-218: Be more careful when reasoning of possible transmission (see the general point above)

L 261: Please provide better justification for selecting these antibiotic. Please explain what you base that they are commonly used on?

L 297: Can you really claim that you have studied the "driving" of ABR, isn't it more the "occurrence" (as the number and designing qualify for saying prevalence) of ABR

Reviewer #2 (Remarks to the Author):

Review report COMMSENV-24-2398-T

This paper describes a longitudinal field study in which pigs and farmers in semi-intensive systems and free-range pig systems in Uganda were sampled to assess the presence of antimicrobial resistance using both phenotypic and genotypic approaches. The aim of this study was to quantify and compare rates of resistance to commonly used antibiotics, to identify factors associated with resistance and to investigate the potential for cross-species transmission of resistance between humans and animals on the monitored farms.

The paper describes a relevant and interesting topic as antimicrobial resistance is a high priority public health problem, especially in low-and-middle income countries where it is expected that intensive livestock systems will expand rapidly in the upcoming years with a related expected rise in antimicrobial use and therefore higher risks for induction and dissemination of antimicrobial resistance. Although the study in itself is relatively well designed, the manuscript as such in its current state lacks scientific quality and clarity to suit publication. I would suggest to ask the authors to fundamentally revise the manuscript in order to make it suitable for publication. I will explain the flaws below.

General remarks

I prefer to use 'antimicrobial' instead of 'antibiotic'. It is a semantic discussion, but in general 'antimicrobial' is more commonly used in scientific literature.

Abstract

Line 36: insert a tab between 'high(22.5%)'

Line 38: specify exactly how much more likely transmission was

Introduction

Introduction is well written

Line 56: I would suggest to add a reference to build the statement about the risk of higher disease burden, as I think that this is not really black and white because in intensive production systems it is generally easier to comply to biosecurity practices.

Line 58: It is not clear where you are referring to with 'the impact of this rise on countries'. Which impact do you refer to?

Line 63: This means that in the rest of the manuscript semi-intensive and peri-urban are both classified as 'semi-intensive'? Is it reasonable to merge them together?

Line 65: Why is it likely that the differences will increase compared to the already existing difference? Is it expected that semi-intensive systems will consume even more antimicrobials in the future?

Line 69-70: Please explain in more detail why you've chosen E. coli and Klebsiella as indicators

Line 71: what kind of differences are you referring to when talking about bacterial, host and production systems?

Results

A major issue in the chosen methodology is that there is not only a difference in farm systems that are being studied, but there also is a spatial difference presence in this study which is not addressed at all. How sure are you that the found differences in ABR profiles between semi-intensive and free-range farm systems in fact are not caused by background ABR in the environmental context? As the country of Mubende and Kampala are approx. 150km apart from each other and most probably the whole context is different (think of water sources, sewage systems, origin of feed and others). This might also be concluded from fig S 1a; also in low exposure persons AMR levels are already higher in environments with semi-intensive farms which could be related to the presence of farms, but also might have other risk factors.

Line 80: Figure 1 is too simplistic in my view. It can give the impression that antibiotic use in farmers and pigs lead to ABR gene carriage in only one bacteria. It would also be better to incorporate 'breed, stocking density, feeding and housing' within the circle of 'pig factors' leading to AMU. I would also like to see some explanation how 'time' is believed to have an effect on antibiotic use in farmers and pigs?

Line 84: please discriminate between numbers of fecal samples in pigs and humans.

Line 92-96: To me this sounds a bit like 'cherry picking'. Here, the presence of a correlation is studied between the number of antimicrobials (out of a total of six) to which an isolate showed phenotypic resistance and the number of copy counts of four different resistance genes (for only 3 different antimicrobial classes where for example tylosin resistance was tested genotypically, but not phenotypically). I don't see a clear biological link between these 2 parameters, except maybe for cross resistance but at least this should be thoroughly discussed. Fig 2 is difficult to interpret. E.g. fig 2a; the proportions summed per production system is far more than 1 (so impossible), furthermore, where do the numbers above the bar stand for? In fig 2b: how is the correlation coefficient and p-value calculated? There are 2 separate lines (one for free-range and one for semi-intensive) but only one R-value and one P-value. Are they merged together? It would also benefit to include the individual observations (dot plot).

Line 98-102: See also my remarks at the methods section. I don't understand why both *E. coli* and *Klebsiella* spp. were isolated and analyzed as one uniform group. It is not clear whether the found *E. coli* and *Klebsiella* were evenly distributed amongst hosts and production systems. A major issue is that they are not separately analyzed whilst having different resistance patterns, which troubles the interpretation of results (especially the figures) as in my view it is impossible to extrapolate these results to the whole family of Enterobacteriaceae.

Line 100: How was 'lower resistance' defined? Figure 3 needs major revisions. It is really not clear where we are looking at. What was the dependent outcome variable in fig 3A? The legend says 'phenotypic resistance' but how was this defined and calculated? Fig 3b: what is adjusted ABR prevalence? The legend mentions trp/s but this is not findable in the figure. Same for c-f; it is really not clear what exactly is being displayed and how to interpret this.

Line 103-104: I don't get fig S1B. It cannot be seen in this figure that references have lower levels of ABR. Furthermore, I don't get the legend 'note that b also reflects the dynamics of exposure references'. Referents were taken both from semi-intensive and free-range farms if I'm correct? Does this figure encompass all samples taken (both pigs, farmers and referents)?

Line 104: 'Compared to ciprofloxacin'. I would suggest to write that the highest rate of resistance was found to ciprofloxacin. Furthermore, where do the OR refer to? It is not useful to calculate the ORs when comparing resistance to different antimicrobials as they are all independent variables.

Line 109: fig S2 in the legend refers to 'Urban', but that can nowhere be found in the figure.

Line 112-113: was the use of antibiotics assessed every visit, or only at the start of the study? This is not clear (see also methods section). It really is an omission that nothing is known about which antibiotics were used (at least in the pigs).

Line 113: What is meant with 'predominantly housed'? That in both systems most pigs lived in confined stables?

Line 116-117: Only one OR is being given, but two different populations (namely humans and pigs) are studied so I would expect two ORs (one for humans and one for pigs) so I would analyze them separately.

Line 125-127: See my remarks at line 98-102

Line 133: Fig S4 is showing results from exposure referents, is not related to the described odds in line 132. Furthermore, how is ABR (fig 3e on the Y-axis) defined? What is the relevance of this association?

Line 135: please also provide confidence intervals when giving ORs.

Line 137: Please find my previous remarks; when was 'antibiotic use' measured and how was this association calculated?

Line 141-143: How was this 'gene transfer' defined and calculated? Was the presence of a particular resistance gene (total of 4) defined at the farmer level and 2 months later at the pig level? In the case the gene was found in both species (human at $t=0$ and pig at $t=0+1$), then it was classified as 'gene transfer'? Why was it not analyzed at the same sampling point? How long is it expected that resistance genes will be present in the gut after transmission?

Line 147: Fig S3 is not about MDR profiles, do you refer to S2 probably? Please rephrase in line 147 "...greatest number of MDR isolates was observed in...". Fig S7 is impossible to interpret in the current layout and legend. How can I see where a possible transmission event has occurred?

Line 143-160: I doubt whether finding of a similar MDR type is equivalent to a transmission event. There also could be a common external source of this MDR type present. Furthermore, some MDR patterns are quite frequently found (see fig S2) so they could also be more or less universally present in the whole population of pigs and farmers? I would suggest to critically review your definition and classification of a 'transmission event'.

Discussion.

The discussion part is relatively short and lacks a more critical review on the chosen methodology and limitations of the study. At least I would like to see a better discussion on the chosen methodology (selection of farms and animals in different geographical and socio-ecological areas, choice of antimicrobials, used laboratory methods, statistical test performed and biological explanation of found correlations). I'm missing the substantiation of the quite bold statements in the conclusion. Specifically:

Line 164: You have not quantified ABU so how can you conclude that ABU was lower in free-range farms in your study?

Line 167: different levels of ABR in *E. coli* vs *Klebsiella* is really not relevant in your study, it only adds to confusion and probable misinterpretation of results so I would delete it here.

Line 169: You observed a small (but significant) rise in ABR levels (however they were defined), could this also be a result of season or other determinants that influenced ABU for instance?

Line 174: what kind of 'externalities' are referred to here? They should be explained. And what kind of 'preparations' are needed? For what purpose?

Line 177-178: I think this statement is too bold. What you have found are differences in ABR patterns between the two distinct farming systems, but there could be many other factors involved that explain this difference apart from the farming system (e.g. level of ABU which is not measured and could be different between the two systems, environmental background). So I would rewrite this sentence.

Line 184: how was 'use of medication' correlated with ABR? As a binary independent variable? Was this medication use in animals or in humans?

Line 187-193: it should be clear that these studies all were about humans (not animals).

Line 195-197: This conclusion cannot be drawn from the results that there was a 'sustained antibiotic pressure', so should be substantiated or rewritten.

Line 200-202: There is not always a direct link between the finding of resistance genes and use of an antimicrobial from the same class. Especially with 'MDR', there is often co-resistance (see e.g. *Frontiers | Co-resistance to Amoxicillin and Tetracycline as an Indicator of Multidrug Resistance in Escherichia coli Isolates From Animals*). So the conclusion that a rise in *tetQ* and *tetB* suggests a higher use of tetracyclines cannot be drawn.

Line 204-211: See my previous remarks about 'transmission events' based on your classification. I doubt whether this is a sound method to measure 'transmission' and should be better substantiated with good references. Even then, a common external source could be well possible.

Line 212-216: The sentence does not read well; what is meant by "high carriage of the *dfrA1* and *tetQ* genes relative to that of their pig"? Line 215 could be red as farmers using *tmp/s* themselves but this is not often used in humans as far as I know (but I'm not familiar with human use in Uganda).

Line 224-228: I don't support these conclusions. The only thing that can be concluded is that higher ABR rates and gene carriage (of selected antimicrobials) have been found on semi-intensive farming systems in both humans and animals. So this needs to be rewritten. Also the statement that the farm is the ideal level for interventions cannot be argued based on your findings and is also not discussed in the discussion part. Finally, I do not see how this approach with MDR-gene load relationships presents diagnostic opportunities for evaluations. It should be better substantiated and explained how you see this.

Materials and Methods

This section is currently of poor scientific quality and clarity. A lot of crucial details are missing in the description of what has been done.

Field study. It is not clear how monitored farms specifically were selected, how animals were selected (only one animal per farm and same animal followed during the whole study?) and how samples were taken (e.g. swab, fresh faecal droppings). How was the 'farmer' defined (as especially in free-range systems there are mostly more people dealing with the animals)? It is also not mentioned how samples were stored and transported (e.g. transport conditions, transport time). It is also not clear how the reference group was selected (line 248-249); based on which criteria? How was defined whether someone was a 'high' or 'low' risk person?

Microbiological analyses. It should at least be summarized which laboratory procedures have been followed to isolate and identify *E. coli* and *Klebsiella*. How was finally one colony be selected for further analysis on ABR? Why was chosen to both include *E. coli* and *Klebsiella* as I might assume that *E. coli* can be found in every faecal sample. Disc diffusion was used, it should be clear from which company these discs were derived and which procedures were followed. Why were only these six antibiotic classes tested (and why both gentamicin and streptomycin) and not others? How and when was 'resistance' measured? Which criteria were used to classify an isolate as 'resistant' (e.g. clinical breakpoints, ECOFFs, CLSI or EUCAST)? Why was chosen to look for gene carriage for only 4 selected genes? Why were these genes chosen and not others?

Line 239: pairs were followed for 10 months I assume (as you have 6 time points and 12 months would require 7 time points).

Line 232: The National Council is from Uganda I suppose? Please make clear.

Line 243-244: it should be clear that 35 pairs per study arm were followed (70 in total)

Line 249: S1 refers to results so it should be referred to in the results section, not here.

Line 253: related to Table S1. Was this information requested to the farmers every visit (I assume only at the start which means that e.g. use of antibiotics is hard to analyze in a longitudinal study)? Why is the row 'antibiotic use' in referents almost empty? How to interpret 'antibiotic use' (Y/N) in the table referring to pigs? Is this antibiotic use in the last 2 weeks before the start of the study? Did it only refer to the animal that was sampled? Was it not monitored during each visit? Why was the type/class of antibiotic used not recorded? Figure S10: I don't really know how to interpret this figure, especially the numbers in the upper part (9 vs 10 samples of men and women? Where does this stand for?). And why is marital status important?

Line 261: seven antibiotics were tested, not six

Line 262-266: This assumption does not hold in my view and is also not really substantiated by Rhouma et al. I would like to see robust evidence for the assumption that AMR profiles in only one selected colony is indicative of AMR profiles in the whole population of Enterobacteriaceae in the gut. I think this is not true as there can be a wide variety of resistance profiles within a population.

Line 266-267: This is not clear. Did you extract an aliquot of a fecal sample? Which methods were used? How was made sure that it contained 'high molecular grade DNA'?

Line 271: Tables S3&4 and Figure S4 should be referred to in the results section. In table S3, the legend talks about six tested antibiotics, but in fact it were 7. It is also not clear where the text on S page 9 (between table S3 and S4) about factors associated with ABR gene belongs to?

Statistical analyses.

Line 302: What does time (T+1 mean? That is 2 months after the sampling moment in the farmer of that animal?

Line 301-303: It is not clear what exactly was modelled. Only gene carriage (and then per each of the 4 different genes)?

Line 303-308: It is not clear how 'transmission' was defined. You are assuming that sharing a multi-drug resistant strain between a pair is a rare event. But how was this analyzed? Was transmission suspected when isolates with exactly the same antimicrobial resistance profile (phenotypically) were found in both farmer and pig? Could it also be that this is not transmission but has a same external source (e.g. drinking water, environment)?

Figures and tables

Many tables and figures must be improved. Often the legend does not clearly correspond with the figure and many figures are hard to interpret, even after reading the legend which means that the legend also should be rewritten to clearly explain the reader how to interpret the figures. For figures where correlations were calculated (e.g. 2b; 3e, 4b and others) I would suggest to draw a dot plot to see the individual observations rather than a straight line indicating the relationship.

Reviewer #3 (Remarks to the Author):

I appreciate the great work put together by Muwonge et al. providing critical information for the long-lasting battle against antibiotic resistance issue. Current research is mostly conducted in high-income countries, while the LMIC should never be left out. The reviewer considered the manuscript is overall clear and well-written. However, the reviewer believes the manuscript can be improved by expanding the materials and method section to clarify the study. Please see the particular suggestions as follows:

Abstract

1. Line 33: Please briefly describe the experimental design. How many systems and free-range pigs were included in the study? How was the "six times" more antibiotic-resistant bacteria determined? What specimens were used for such determination?

2. Line 36 – 37: Please elaborate what the percentages indicated. Does it mean ABR bacteria was isolated in 22.5% of the farmers and pigs in semi-intensive systems?

3. Line 37: Please provide the length of the study. Any reason for the increase? Was it because more antibiotics were used by the farmers?

4. Line 38: Transmission of ABR bacteria? Needs to be clarified.

5. Line 38: Please provide the exact number of times instead of an approximation (~3).

Introduction

1. The reviewer appreciates the well-written introduction as it is concise and spot-on.

2. Line 61 – 62: Reference needed for the statement for higher ABR risk for LMIC.

Materials and method

1. Line 239: ... were followed "bimonthly" for a year (six time points) ...

2. Line 238 – 253: Please provide further description for the production systems and households. What was the average number of pigs? Was there a treatment protocol available for farmers?

3. Line 282: Please avoid using abbreviations that are not previously defined in the manuscript. For example, "qqnorm" should be quantile-quantile normality plot.

4. Line 274 – 284: The reviewer suggests further clarification on the model construction process.

1. Potential confounders should be evaluated and included for each model. For example, the ABU of farmers and pigs should be included in Model 1 as a confounder. I suggest using directed acyclic graphs (DAG, available at <https://www.dagitty.net/>) to identify potential confounders, and assess their effects to the outcome - exposure association using the model. Unable to control for confounders can lead to biased result interpretation and conclusion.

2. Many assumptions described here only apply to linear regression models (Gaussian). Assumptions for binomial logistic regression models should be assessed and reported.

3. How are the random effects modeled? Were they modeled as random intercepts, random slopes, or both? Were they modeled separately, crossed, or nested?

4. Please make sure the variable names are consistent throughout the manuscript. For model 1, setting is used in the fixed effect column but changed to Production System in the interaction column. Also, the random effect "visit" is described as "time" in the footnote.

5. Line 285 – 289: It is not clear why a logistic regression model was used. If the ABR prevalence was the outcome, a Poisson regression model with the denominator of the prevalence (proportion) as the offset would be appropriate. In fact, Table 2 states "an isolate expressing resistance or not" was modeled as the outcome, which is in conflict with the in-text description.

6. Line 295 – 296: Please provide the normalization formula or a reference to ensure reproducibility.

7. Line 294 – 295: Please describe how the trend of copy counts modeled?

8. Line 298: It seems Model 4 is just Model 2 with more fixed effects added? If so, why report both? Also, I highly recommend adding an "outcome" column to Table 2.

9. Line 303 – 308: It is not clear why the farmers and pigs “not” from the same household were modeled. Was there a logical reason for household-to-household transmission for farmers and pigs?

10. Overall, I suggest the authors expand the statistical analyses section due to the number of models being reported. Each model must be explicitly reported, including the outcome, fixed effect, random effect, and the type of model. The current form requires the reader to constantly go back and forth to Table 2, which is not ideal in terms of reading flow.

Results

1. Line 79 – 96: This section should be moved to M&M.

2. Figure 3a: The reference level of each variable should be added either to the figure or in a footnote.

Communications Earth & Environment is committed to improving transparency in authorship. As part of our efforts in this direction, we are now requesting that all authors identified as ‘corresponding author’ create and link their Open Researcher and Contributor Identifier (ORCID) with their account on the Manuscript Tracking System prior to acceptance. ORCID helps the scientific community achieve unambiguous attribution of all scholarly contributions. You can create and link your ORCID from the home page of the Manuscript Tracking System by clicking on ‘Modify my Springer Nature account’ and following the instructions in the link below. Please also inform all co-authors that they can add their ORCIDs to their accounts and that they must do so prior to acceptance.

Version 1:

Decision Letter:

Dear Dr Muwonge,

Your manuscript titled "Pig production system drivers of antibiotic resistance in Uganda" has now been seen by our reviewers, whose comments appear below. In light of their advice we are delighted to say that we are happy, in principle, to publish a suitably revised version in Communications Earth & Environment.

We therefore invite you to revise your paper one last time to address the remaining concerns of our reviewers. At the same time we ask that you edit your manuscript to comply with our format requirements and to maximise the accessibility and therefore the impact of your work.

EDITORIAL REQUESTS:

*****Please take care to match our formatting and policy requirements. We will check revised manuscript and return manuscripts that do not comply. Such requests will lead to delays. *****

SUBMISSION INFORMATION:

OPEN ACCESS:

Communications Earth & Environment is a fully open access journal. Articles are made freely accessible on publication. For further information about article processing charges, open access funding, and advice and support from Nature Research, please visit <https://www.nature.com/commsenv/open-access>

Link Redacted

Best regards,

Mengjie Wang

Associate Editor, Communications Earth & Environment

<https://www.nature.com/commsenv/>

Consulting Editor, Communications Sustainability

<https://www.nature.com/commssustain/>

Bluesky: @commsearth.nature.social; @commssustain.nature.com

REVIEWERS' COMMENTS:

Reviewer #1 (Remarks to the Author):

The revised Ms has improved a lot. Still there is one conceptual concern that must be adjusted before a potential publication (the first one below), and some editorial comments as well.

L 41 To refer to shared resistance (phenotypic or genes) as transmission or transmission events is misleading and not scientific sound. Please use the wording potential transmission events throughout the MS and not only in the discussion.

L304 missing the reference

L 308-309, isn't "interaction between" more precise than "influence"

L419-421 some odd wordings here, please check!

L 482 "Livestock use"?

L 488 AMR and MDR aren't two different entities justifying "and", please reword.

L 489 (b) this statement is confusing – please be more precise

Reviewer #2 (Remarks to the Author):

The manuscript has been greatly improved compared to the previous version.

A few comments:

-Carefully check all the grammar/spelling and the insertion of proper references

-Change ABU into AMU where applicable (also in figures)

I don't have enough time at the moment to critically review all the changes that have been made (and unfortunately not all comments can be traced back in the manuscript) but I believe that the authors have made an extensive effort to revise the manuscript and is now ready for acceptance.

Reviewer #3 (Remarks to the Author):

Dear authors,

I really appreciate your efforts on responding to my comments and suggestions. The manuscript has been substantially improved in every aspect. Just a few minor comments:

1. Line 46 - 47: The OR and stats outcomes of the last item (copy number of tetQ) is missing in the abstract.

2. For my question regarding average number of pigs and treatment protocol, I was considering the farm characteristics rather than farmers', which is being summarized in Table S1. Does the data about the number of pigs that each farmer was raising and whether farmers had a protocol to follow for AMU available?

Looking forward to the revised manuscript!

Query	Reviewer comments	Author Response
Reviewer 1	This study is not unique - comparisons of AB-burden among farmers and pigs between different pig farming systems has been conducted in other low/middle income countries (e.g. Lunha et al. “Antimicrobial Resistance in Fecal Escherichia coli from Humans and Pigs at Farms at Different Levels of Intensification.” Antibiotics (Basel, Switzerland) vol. 9,10 662. 30 Sep. 2020, doi:10.3390/antibiotics9100662. , Hallenberg et al. “Antibiotic use in pig farms at different levels of intensification-Farmers' practices in northeastern Thailand.” PloS one vol. 15,12 e0243099. 11 Dec. 2020, doi:10.1371/journal.pone.0243099; Hickman, et al. “Exploring the Antibiotic Resistance Burden in Livestock, Livestock Handlers and Their Non-Livestock Handling Contacts: A One Health Perspective.” Frontiers in microbiology vol. 12 651461. 20 Apr. 2021, doi:10.3389/fmicb.2021.651461). However, despite this lack of novelty the paper provides relevant information about the situation in Uganda, stressing the importance to manage the intensification of pig farming systems in a sensible way.	We thank the reviewer for their comments and for highlighting relevant literature on similar population-level studies. These however are predominantly from Asia, particularly Thailand. The lack of African studies, mean that there is a novelty in our work. Our study included Swine and non-swine exposed reference groups, the key distinction lies in the depth provided by our one-year longitudinal study following 70 farmers, 35 in each stratified within the setting. This generated 877 sample and 668 sentinel bacteria offering robust statistical power. We therefore argue there is novelty here in the following a) Design,- Longitudinal(temporal scale), stratified between and within two production systems. b) Geography- Our literature review for Africa, shows this is the only study to date at this scale documenting the risk of a rapidly growing sector developing pig industry. c)Depth- We use two sentinel bacteria phenotypes and contrast them with ARG carriage in their resident microbiomes, revealing a strong signal that forms the basis for transmission inferences.
	One conceptual flaw of the paper is that the authors use sharing of antibiotic profile (different aspects) in pigs and humans as a proxy for transmission. This might be, but there is not solid scientific evidence behind this. So please revise throughout the ms say “sharing” instead of “transmission”. You may, of course, speculate that in the discussion that transmission occurs	In this study we assumed a) relationship between phenotypic and genotypic resistance(as reported elsewhere);, and (b) a correlation at the pathogen and microbiome levels across scales. After we show this correlation, it is what formed the basis for using phenotypes as the unit for investigating transmission. We however acknowledge that the findings from this analysis require further validation using more granular molecular tools.
	L: 35-37, it is confusing to which percentage refers to what? Please clarify	The entire abstract has been revised for clarity [page 2 line 30-46]
	L 46: There are substantial amounts of publications and strategies on control strategies for ABR in livestock – please revise.	This has been revised as advised [page 2 line 52-53]
	L 48: Is a publication from 2018 really recent? There must be more recent publications supporting this very general statement.	More recent publication have been cited [page 2 line -40]
	L 51: Usually increased incomes are added to this list of factors – likely more important than general migration.	This has been added as advised [page 2 line 55-56]
	L56-57: Please not that increased intensification/productivity not necessarily have to be followed by	We accept this nuance however this premised on the LMIC context

	increased ABU us as prophylaxis (e.g. Denmark). Please nuance.	
	L 102: do the 23% refers to pigs and humans as well.	This has now been revised, but for clarity it compares the farming, and swine exposure references. We have also added figures in supplementary Fig S3
	L 104: please provide reference group data here as well. Also, the reference groups seems to be very heterogenous (L248-249)	We have now adopted the swine exposed(SE) and non-swine exposed group(NSE group) We have also added figures in supplementary Fig S3
	L 104-L109: Over all, or from humans in semi-intensive, pigs in free-ranging etc? Please clarify also in the running text here.	This has been clarified as requested, the percentage is for farmers, and comes from Fig 2c[Line 103-104 page3]
	L 112: “pig has used antibiotics” sounds odd, please rewrite L 128: Same as L 104	This has been clarified as requested Line 114 page 4
	L 145: the common definition of MDR is that the bacteria is resistant to antibiotics of 3 different classes – please change and check data.	This has been amended in [line 133 page 5]
	L 166: it is obvious that production systems are associated with production systems, pls rewrite.	This has now been edited on [line 171-172 page 6]
	L172: “ABR risk”, is very much jargon. Please be more elaborate.	This has now been edited on [line 296 page 10]
	L 173-74: , “likely due to..” statement is very vague/general and doesn’t add to the discussion, please omit.	This has now been edited on [line 191 page 6]
	L 177-78: Highly speculative and general, not supported by the data. Pls omit.	This has now been edited on [line 174-179 page 6]
	L 182-184: This is contradictory and confusing. Pls rewrite.	This has now been rewritten as advised line
	L 185: “This finding”, please specify.	This has now been clarified
	L 187-L192: Please shorten this section -it is hard to follow the reasoning.	This section has now been revised line 191-205 page 6
	L 198: is 2017 recent? Please look for more recent reviews.	This has now been revised a more recent reference cited on line 57 page 2
	L 203: Reference “27” is an odd one?! Please delete.	We have now edited the reference to include the details including the URL https://stud.epsilon.slu.se/16132/1/selling_k_200128.pdf
	L 213-218: Be more carefull when reasoning of possible transmission (see the general point above)	In this study we assumed a) relationship between phenotypic and genotypic resistance(as reported elsewhere):, and (b) a correlation at the pathogen and microbiome levels across scales. After we show this correlation, it is what formed the basis for using phenotypes as the unit for investigating transmission. We however acknowledge that the findings from this analysis require further validation using more granular molecular tools. This one of the things recommended for future work on line 192 page 7
	L 261: Please provide better justification for selecting these antibiotic. Please explain what you base that they are commonly used on?	This is because antibiotics were selected based on their utility in both systems, and, as you know, Tylosin is exclusively a veterinary drug. Given that we are using qPCR to track gene abundance, the selection had to be both clinically relevant and focused on genes routinely present. It so happens that ermB is routinely present,

		allowing us to speculate about Tylosin use and resistance. Line 350-3 page 11
	L 297: Can you really claim that you have studied the “driving” of ABR, isn’t it more the “occurrence” (as the number and designing qualify for saying prevalence) of ABR	The study was powered to estimate AMR prevalence over the 1 year period, then use the ~ 870 dataset we identify factors(“Drivers”) associated with AMR. This provides an estimate of prevalence in this group, with the observed temporal increase comparable to the recent national estimate. Line 199 page 7
Reviewer 3	I appreciate the great work put together by Muwonge et al. providing critical information for the long-lasting battle against antibiotic resistance issue. Current research is mostly conducted in high-income countries, while the LMIC should never be left out. The reviewer considered the manuscript is overall clear and well-written. However, the reviewer believes the manuscript can be improved by expanding the materials and method section to clarify the study. Please see the particular suggestions as follows:	The authors thank reviewer for these observations
	1. Line 33: Please briefly describe the experimental design. How many systems and free-range pigs were included in the study? How was the “six times” more antibiotic-resistant bacteria determined? What specimens were used for such determination?	Given the limited text allowance for abstract, limited methods were included in that section. We have added some clarity in abstract within allowable text limits Page 1 line 31-36 . We have also elaborated this in the methods section- Design on page 10 line 310 . The prevalence estimated is model output in figure 2b. We note an error here, Pigs in Semi-intensive 23.8% and Free range 7.5%, The prevalence in Semi-intensive is 3 times that of free range. From the model Semi intensive pigs are 1.4 times more likely to carry resistant E.coli
	Line 36 – 37: Please elaborate what the percentages indicated. Does it mean ABR bacteria was isolated in 22.5% of the farmers and pigs in semi-intensive systems?	Please note that this section of the abstract has been revised. The average percentages of antibiotic-resistant bacteria isolated from farmers or pigs within a given system are model outputs fig 2c, representing adjusted prevalence/proportions of resistance.
	Line 37: Please provide the length of the study. Any reason for the increase? Was it because more antibiotics were used by the farmers?	This was a year-long longitudinal study that involved monitoring farmers and their pigs, with visits conducted every two months. We believe this likely reflects an increasing trend in resistance, as similar patterns have been reported in other hosts within the same country(REF).
	Line 38: Please provide the exact number of times instead of an approximation (~3).	This has now been edited as requested by the reviewer on [page 2 line 38]of the abstract
	Introduction 1. The reviewer appreciates the well-written introduction as it is concise and spot-on.	We appreciate the positive feed back
	. Line 61 – 62: Reference needed for the statement for higher ABR risk for LMIC.	This references have now been added, see page 3 line 62
	Materials and method 1. Line 239: ... were followed “bimonthly” for a year (six time points) ...	This has now been edit as recommended by the reviewer

	Line 238 – 253: Please provide further description for the production systems and households. What was the average number of pigs? Was there a treatment protocol available for farmers?	We have provided an elaborated picture of the typical semi-intensive and Free-range farmer in the supplementary and this is well cited in the manuscript. We have done so because of text limitation Page 5 line 112-122
	Line 282: Please avoid using abbreviations that are not previously defined in the manuscript. For example, “qqnorm” should be quantile-quantile normality plot.	The abbreviations have been revised to include their full forms, as advised page 13
	Line 274 – 284: The reviewer suggests further clarification on the model construction process.	The model construction and selection approach has now been added Page 12 line 378-381
	Potential confounders should be evaluated and included for each model. For example, the ABU of farmers and pigs should be included in Model 1 as a confounder. I suggest using directed acyclic graphs (DAG, available at https://www.dagitty.net/) to identify potential confounders, and assess their effects to the outcome - exposure association using the model.	We thank the reviewer, for this suggestion, the potentially confounding factors were included in the initial model but the backward selection process eliminated them. None the less we have used the DAGitty to evaluate co-founding effects and minimise bias. We have updated our conceptual framework to reflect the directionality of effects. We have then use time and house hold(production cycles) as random variable. Instead, we used the Conditional Inference Tree, an acyclic tree approach for determining the structure and significance of variables in models. The results highlight the most important variables depending on the model (1–4). See Supplementary Page 13, Lines 347 & 371.
	Unable to control for confounders can lead to biased result interpretation and conclusion.	Yes we have revised our conclusion, to reflect this limitation see page 2 line 45
	Many assumptions described here only apply to linear regression models (Gaussian). Assumptions for binomial logistic regression models should be assessed and reported.	Just to clarify here we used the Generalised linear regression models with a binomial link. We only use Gaussian for the modelling of gene carriage. Therefore standard assumption such as Independence and no-multicollinearity were ensured. We ensure independence by accounting for clustering using fixed and random effects model. Page 13 line 370 & 391
	How are the random effects modelled? Were they modelled as random intercepts, random slopes, or both? Were they modelled separately, crossed, or nested?	The random effects were modelled as intercepts, with the Farm ID nested within a sampling time point. This has been clarified in on page 10 and line 388-391
	Please make sure the variable names are consistent throughout the manuscript. For model 1, setting is used in the fixed effect column but changed to Production System in the interaction column. Also, the random effect “visit” is described as “time” in the footnote.	This has been edited both in the manuscript and the table 2
	Line 285 – 289: It is not clear why a logistic regression model was used. If the ABR prevalence was the outcome, a Poisson regression model with the denominator of the prevalence (proportion) as the offset would be appropriate. In fact, Table 2 states “an isolate expressing resistance or not” was modeled as the outcome, which is in conflict with the in-text description.	We appreciate the reviewers' suggestion to model the proportion of resistance. However, we note a few challenges with this approach: A) The maximum denominator per household would be 4, based on the number of sentinel bacteria recovered per household. As a result, the possible proportions would be limited to 0.25, 0.50, 0.75, and 1.00. B) Not all hosts will have bacteria recovered, particularly Klebsiella, which would further shrink the possible

		proportions, all in all this would restrict the variability available for modelling using a Poisson model. Given these limitations, a random-effects logistic regression model would be a more suitable alternative.. None the less we have explored this your proposal and .
	Line 295 – 296: Please provide the normalization formula or a reference to ensure reproducibility.	This has now been added see page 13 line 389
	Line 294 – 295: Please describe how the trend of copy counts modeled?	We use Generalised Linear regression model, with the outcome as the normalised gene count as shown in Model 2 in table, Figure 2e
	Line 298: It seems Model 4 is just Model 2 with more fixed effects added? If so, why report both? Also, I highly recommend adding an “outcome” column to Table 2.	The outcome variable for each model has now been clarified in Table 2. Models 2 and 4 are distinct but address the same questions using genotypic and phenotypic data, respectively. Specifically, Model 2 is a linear model analysing AMR gene carriage, while Model 4 is a logistic model examining the AMR phenotype.
	. Line 303 – 308: It is not clear why the farmers and pigs “not” from the same household were modeled. Was there a logical reason for household-to-household transmission for farmers and pigs?	This analysis is based on our assumptions on transmission events as
	Overall, I suggest the authors expand the statistical analyses section due to the number of models being reported. Each model must be explicitly reported, including the outcome, fixed effect, random effect, and the type of model. The current form requires the reader to constantly go back and forth to Table 2, which is not ideal in terms of reading flow.	We have added a paragraph in the Methods section that explains each model and its corresponding outcomes. In the Results section, we specify the model from which each result is derived. See page 10-13
	Results 1. Line 79 – 96: This section should be moved to M&M. 2. Figure 3a: The reference level of each variable should be added either to the figure or in a footnote.	We have moved this section a suggested by the reviewer- See page 10 line 309-317
Reviewer 2	This paper describes a longitudinal field study in which pigs and farmers in semi-intensive systems and free-range pig systems in Uganda were sampled to assess the presence of antimicrobial resistance using both phenotypic and genotypic approaches. The aim of this study was to quantify and compare rates of resistance to commonly used antibiotics, to identify factors associated with resistance and to investigate the potential for cross-species transmission of resistance between humans and animals on the monitored	We appreciate the reviewer’s feedback and will revise the manuscript to enhance its quality.

	farms. The paper describes a relevant and interesting topic as antimicrobial resistance is a high priority public health problem, especially in low-and-middle income countries where it is expected that intensive livestock systems will expand rapidly in the upcoming years with a related expected rise in antimicrobial use and therefore higher risks for induction and dissemination of antimicrobial resistance. Although the study in itself is relatively well designed, the manuscript as such in its current state lacks scientific quality and clarity to suit publication. I would suggest to ask the authors to fundamentally revise the manuscript in order to make it suitable for publication. I will explain the flaws below.	
	General remarks I prefer to use ‘antimicrobial’ instead of ‘antibiotic’. It is a semantic discussion, but in general ‘antimicrobial’ is more commonly used in scientific literature.	This has now been updated in the text
	Abstract Line 36: insert a tab between ‘high(22.5%)’ Line 38: specify exactly how much more likely transmission was	The abstract has now been edited accordingly see page 1 line 34-43
	Line 56: I would suggest to add a reference to build the statement about the risk of higher disease burden, as I think that this is not really black and white because in intensive production systems it is generally easier to comply to biosecurity practices.	This reference has now been added on line.. page 2 line 51 . There is indeed a reported increased risk of infectious diseases, particularly zoonotic diseases. While biosecurity measures are expected to improve, they are often weighed against profit margins in LMICs.
	Line 58: It is not clear where you are referring to with ‘the impact of this rise on countries’. Which impact do you refer to?	Here, we refer to the impact of fold change in AMU, which has now been clarified on page 3 line 64-67
	Line 63: This means that in the rest of the manuscript semi-	We have used peri-urban as the proxy for semi-intensive, for most countries in the Africa and especially east

	intensive and peri-urban are both classified as ‘semi-intensive’? Is it reasonable to merge them together?	African region that holds true. More animals reared closer to the market, with many on small piece land, housed, fed on improved feed, using AI for breeding, using veterinary extension services, and participating for in agriculture with aim of profits be that as supplementation of house hold incomes
	Line 65: Why is it likely that the differences will increase compared to the already existing difference? Is it expected that semi-intensive systems will consume even more antimicrobials in the future?	In pig production, antibiotic use is predicted to increase threefold globally[Page 3 line 65]. This is partly due to anticipated intensification, which brings an increased risk of infection, especially in regions with high pathogen prevalence and dense stocking densities. This is expected to lead to greater antibiotic use for prophylaxis, therapeutics, and growth promotion, with the latter likely contributing a smaller component.
	Line 69-70: Please explain in more detail why you’ve chosen E. coli and Klebsiella as indicators	Global clinical health perspective: E. coli and Klebsiella species are part of the WHO priority pathogen list and are classified as ESKAPE pathogens—highly virulent species with a high risk of antibiotic resistance. E. coli is responsible for one in five foodborne infections, particularly from food of animal origin. Microbiologically, both belong to the Enterobacteriaceae family, but their recovery rates differ in faecal samples, with E. coli at <90% and Klebsiella around 40%. While Klebsiella species often cause respiratory infections, E. coli is primarily associated with the gastrointestinal tract, though both can cause sepsis. The use of antibiotics for these pathogens may vary between farmers and pigs, and it is this difference that we aimed to explore.
	Line 71: what kind of differences are you referring to when talking about bacterial, host and production systems?	Here, we refer to differences in AMR. We aim to examine associations linked to these factors, as they can contribute to designing a tailored response.[page 3 line 79]
	Results A major issue in the chosen methodology is that there is not only a difference in farm systems that are being studied, but there also is a spatial difference presence in this study which is not addressed at all. How sure are you that the found differences in ABR profiles between semi-intensive and free-range farm systems in fact are not caused by background ABR in the environmental context? As the country of Mubende and Kampala are approx. 150km apart from each other and most probably the whole context is different (think of water sources, sewage systems, origin of feed and	We agree with the issue raised by the reviewer. Our assumptions capture this, as we use location as a proxy for production system. However each location is tied to a production system. We acknowledge that this limits our ability to disentangle place from production[Fig 5] We have now included a new figure, S1, comparing swine-exposed (SE) and non-swine-exposed (NSE) reference groups. This reinforces the setting differences independent of pairing between SE and NSE[Extended Data Table S3 & Fig S3:]. Our findings suggest two key points: (a) Although SE individuals are not directly on farms, their AMR profiles—whether mono-resistant or MDR—validate the exposure assumption. (b) NSE individuals (No participants in this production system) exhibit higher AMR levels across antibiotics in peri-urban settings compared to farmers (Figure S3). This suggests that both production system and geographical location may influence AMR patterns, though we are currently unable to disentangle their individual effects.

	others). This might also be concluded from fig S 1a; also in low exposure persons AMR levels are already higher in environments with semi-intensive farms which could be related to the presence of farms, but also might have other risk factors.	
	Line 80: Figure 1 is too simplistic in my view. It can give the impression that antibiotic use in farmers and pigs lead to ABR gene carriage in only one bacteria. It would also be better to incorporate ‘breed, stocking density, feeding and housing’ within the circle of ‘pig factors’ leading to AMU. I would also like to see some explanation how ‘time’ is believed to have an effect on antibiotic use in farmers and pigs?	We have revised this figure to better reflect the underlying complexity. It is important to note the various levels represented. The phenotype is based on two bacteria, while the genes represent the gut microbiome carrying these genes. In this context, the phenotype serves as a sentinel of the gut microbial population, which is why we chose to examine it using these two bacteria.
	Line 84: please discriminate between numbers of fecal samples in pigs and humans.	This has been clarified and this section has now been moved to the materials and methods on page 11 Line 329-339
	Line 92-96: To me this sounds a bit like ‘cherry picking’. Here, the presence of a correlation is studied between the number of antimicrobials (out of a total of six) to which an isolate showed phenotypic resistance and the number of copy counts of four different resistance genes (for only 3 different antimicrobial classes where for example tylosin resistance was tested genotypically, but not phenotypically). I don’t see a clear biological link between these 2 parameters, except maybe for cross resistance but at least this should be thoroughly discussed. Fig 2 is difficult to interpret. E.g. fig 2a; the proportions summed per production system is far more than 1 (so impossible), furthermore, where do the numbers above the bar stand for? In fig 2b: how is the	The assumptions outlined in the causal framework (Figure 6) have guided our analysis. We have specifically clarified the levels, context, host characteristics, the gut microbiome carrying the genes, and the two sentinel bacteria providing the phenotype. In our analysis, we accounted for clustering by time and farm, which inherently captures both geographical effects and temporal farm-specific characteristics. We tested genes encoding resistance to Tylosin but did not examine the phenotype for this antibiotic. This is because antibiotics were selected based on their utility in both systems, and, as you know, Tylosin is exclusively a veterinary drug. Given that we are using qPCR to track gene abundance, the selection had to be both clinically relevant and focused on genes routinely present. It so happens that ermB is routinely present, allowing us to speculate about Tylosin use and resistance.

	correlation coefficient and p-value calculated? There are 2 separate lines (one for free-range and one for semi-intensive) but only one R-value and one P-value. Are they merged together? It would also benefit to include the individual observations (dot plot).	
	Line 98-102: See also my remarks at the methods section. I don't understand why both E. coli and Klebsiella spp. were isolated and analyzed as one uniform group. It is not clear whether the found E. coli and Klebsiella were evenly distributed amongst hosts and production systems. A major issue is that they are not separately analyzed whilst having different resistance patterns, which troubles the interpretation of results (especially the figures) as in my view it is impossible to extrapolate these results to the whole family of Enterobacteriaceae.	Our assumption is that if we detect a correlation between phenotype and genotype, and given that E. coli accounts for only 0.02% of the gut microbiome composition, this phenotype is likely universal to the microbiome or at least specific to the Enterobacteriaceae family. We have highlighted the significance of these two pathogens in the global burden of GIT and respiratory infections. In the same response, we acknowledge their inherent biological and epidemiological differences. As such, we expect them to provide slightly different perspectives on resistance. In Extended Data Figure S3, we show the differences in level of resistance to individual antibiotics, with Klebsiella exhibiting low AMR across selected antibiotic, with differences with Pairs more evident. Given the distribution of MDR, the fitness cost of MDR is likely here for Klebsiella Our current model assumes all bacteria in the same gut environment, exhibiting the phenotypic resistance to individual We have rerun the assuming both bacteria in the same environment but modelled resistance to an antibiotic separately (Extended Data Table S4) We have also modelled both Bacteria per antibiotic and the models are reported in (Extended Data Table S 4 & 5). The individual Bacteria model were attempted but produced less reliable R² We agree that the family of Enterobacteriaceae is diverse. Based on the assumptions in Figure 5, we speculate that other members of this family are likely to reflect the broader differences in AMR observed across production systems, time, and host.
	Line 100: How was 'lower resistance' defined? Figure 3 needs major revisions. It is really not clear where we are looking at. What was the dependent outcome variable in fig 3A? The legend says 'phenotypic resistance' but	Now Figure 2a presents a mixed-effects logistic regression model with a binary phenotypic outcome of "resistance" and "susceptible," as clarified in Table 1. The legend refers to "phenotypic resistance" for clarity. The "adjusted resistance" represents the prevalence based on the model, accounting for all the factors included. For aesthetic purposes, we used the abbreviation TMPs, which

	how was this defined and calculated? Fig 3b: what is adjusted ABR prevalence? The legend mentions tmp/s but this is not findable in the figure. Same for c-f; it is really not clear what exactly is being displayed and how to interpret this.	stands for Trimethoprim and Sulphamethoxazole, and this is mentioned in the table header/footnote.
	Line 103-104: I don't get fig S1B. It cannot be seen in this figure that references have lower levels of ABR. Furthermore, I don't get the legend 'note that b also reflects the dynamics of exposure references'. Referents were taken both from semi-intensive and free-range farms if I'm correct? Does this figure encompass all samples taken (both pigs, farmers and referents)?	This figure previously included only the references. We have now added the pig-farmer pairs Extended Data Figure S3. To clarify, individuals classified under 'Non-swine Exposure' have no contact with pigs due to religious reasons. In contrast, those under 'Swine Exposure' come into direct contact with swine gut content, particularly during the evisceration process in pig abattoirs.
	Line 104: 'Compared to ciprofloxacin'. I would suggest to write that the highest rate of resistance was found to ciprofloxacin. Furthermore, where do the OR refer to? It is not useful to calculate the ORs when comparing resistance to different antimicrobials as they are all independent variables.	We acknowledge that each antibiotic may exhibit varying levels of resistance for a given bacterium. These variables were included in the model, and the odds ratios (OR) presented here are derived from it. We have now included this information in the figure footnote, and both the model and the full model table(Extended Data Table S2) are cited.
	Line 109: fig S2 in the legend refers to 'Urban', but that can nowhere be found in the figure.	We have clarified this in the supplementary Extended Data Figure S3, it is the peri-urban studied and not the urban
	Line 112-113: was the use of antibiotics assessed every visit, or only at the start of the study? This is not clear (see also methods section). It really is an omission that nothing is known about which antibiotics were used (at least in the pigs).	Yes, this was assessed during each visit. The question asked was whether the farmer had used antibiotics on themselves or on their pigs in the two weeks prior to our visit.
	Line 113: What is meant with 'predominantly housed'? That in both systems most pigs lived in confined stables?	This is not the case in Uganda. Some pigs are housed, tethered or free to roam. Housing was predominantly observed in semi-intensive/peri-urban pigs
	Line 116-117: Only one OR is being given, but two different populations (namely humans	The emphasis was to show an overall statistical difference(Figure 2c).

	and pigs) are studied so I would expect two ORs (one for humans and one for pigs) so I would analyze them separately.	
	Line 125-127: See my remarks at line 98-102	This has been noted and responded to accordingly
	Line 133: Fig S4 is showing results from exposure referents, is not related to the described odds in line 132. Furthermore, how is ABR (fig 3e on the Y-axis) defined? What is the relevance of this association?	This is also an error, it supposed to be adjusted prevalence, these are obtained from model 4, Table 2 . This has now been edited
	Line 135: please also provide confidence intervals when giving ORs.	The confidence intervals have now been added throughout the manuscript
	Line 137: Please find my previous remarks; when was 'antibiotic use' measured and how was this association calculated?	Yes this was assess on each visit, the question asked was if a farmer used antibiotics on themselves or on their pig two weeks prior to our visit
	Line 141-143: How was this 'gene transfer' defined and calculated? Was the presence of a particular resistance gene (total of 4) defined at the farmer level and 2 months later at the pig level? In the case the gene was found in both species (human at t=0 and pig at t=0+1), then it was classified as 'gene transfer'? Why was it not analyzed at the same sampling point? How long is it expected that resistance genes will be present in the gut after transmission?	This analysis examines whether the farmer's carriage of any given gene is influenced by that of the pig, and vice versa (Model 3). This inherently requires a lag between the farmer and pig, which we accounted for by using the visits as the lag. For example, we investigated whether the farmer's carriage of a gene at visit T+1 is associated with their pig's carriage of the same gene at visit T. (Extended Data Table S7) It is important to note that studies have shown that shifts in the diversity of a resistome can persist well beyond 7 weeks.
	Line 147: Fig S3 is not about MDR profiles, do you refer to S2 probably? Please rephrase in line 147 "...greatest number of MDR isolates was observed in...". Fig S7 is impossible to interpret in the current layout and legend. How can I see where a possible transmission event has occurred?	Yes indeed, Fig S3 is now Extended Data Figure S4 is about AMR and the temporal scale (visits). This was in reference to S2 and this has been revised. We have now removed Fig S7.
	Line 143-160: I doubt whether finding of a similar MDR type is equivalent to a transmission event. There also could be a common external source of	We have made the following valid arguments as the basis of this analysis a) We assumed that sharing a multi-drug-resistant strain between a pair is a rare event, but if it to

	this MDR type present. Furthermore, some MDR patterns are quite frequently found (see fig S2) so they could also be more or less universally present in the whole population of pigs and farmers? I would suggest to critically review your definition and classification of a ‘transmission event’.	happen I will most likely happen at farm level, and therefore, when detected represents a potential transmission event. So this analysis sought to demonstrate that the probability of an event will be higher at Farm level, then within a subcounty and least between districts/production settings. It is on the basis of this empirical observation that we make inferences on potential transmission events. We acknowledge that certain patterns could be wide spread, but here we show odds of detecting the same pattern shared by a farmer and their pig was higher on the same farm and temporal window than between production /settings
	The discussion part is relatively short and lacks a more critical review on the chosen methodology and limitations of the study. At least I would like to see a better discussion on the chosen methodology (selection of farms and animals in different geographical and socio-ecological areas, choice of antimicrobials, used laboratory methods, statistical test performed and biological explanation of found correlations). I’m missing the substantiation of the quite bold statements in the conclusion. Specifically: Line 164: You have not quantified ABU so how can you conclude that ABU was lower in free-range farms in your study?	We have now expanded the discussion to reflect (From page 6) A) Choice of study design, b) comparison at Microbiome and sentinel bacteria, c) Choice of bacteria, antibiotic and genes, d) The difference in AMR and Gene carriage, The potential effect of AMU and the limitations here, e) Pathogen level differences, f) Host level difference g) Production/setting level difference, i) Inferences on potential transmission inferences
	Line 167: different levels of ABR in E. coli vs Klebsiella is really not relevant in your study, it only adds to confusion and probable misinterpretation of results so I would delete it here.	The sentinel selection is integral to the study design which sought to show the trends based on these bacteria recovered from the same gut. We have also analysed these separately for clarity (Extended Data Table S5).
	Line 169: You observed a small (but significant) rise in ABR levels (however they were defined), could this also be a result of season or other determinants that influenced ABU for instance?	We have examined the effects of seasonality in the models for both sentinel bacteria and individual antibiotics. There is a consistent difference in resistance levels at each visit, particularly for Streptomycin [Extended Data Table S5]. We also see this with Tetracycline, Streptomycin and Gentamycin
	Line 174: what kind of ‘externalities’ are referred to here? They should be	This has been revised/rewritten one page 7. The authors were speculating about the transition from free-range to

	explained. And what kind of 'preparations' are needed? For what purpose?	intensive farming and the associated climate and ecological impacts.
	Line 177-178: I think this statement is too bold. What you have found are differences in ABR patterns between the two distinct farming systems, but there could be many other factors involved that explain this difference apart from the farming system (e.g. level of ABU which is not measured and could be different between the two systems, environmental background). So I would rewrite this sentence.	We have now revised this text to especially the discussion. We acknowledge with current dataset we cannot adequately delineate between geography and production system [line 290, page 10]
	Line 184: how was 'use of medication' correlated with ABR? As a binary independent variable? Was this medication use in animals or in humans?	We asked farmers on each visit whether they had used any medication in the two weeks prior to our visit, with a simple "yes" or "no" response. This served as our proxy for antimicrobial use (AMU) at the farm level. We aimed to strike a balance, ensuring that we did not burden farmers with lengthy questionnaires.
	Line 187-193: it should be clear that these studies all were about humans (not animals).	Yes, these were about Human as few animal studies have longitudinally explored AMR dynamics of a gut sentinel bacteria and the gene carriage of their resident microbiome
	Line 195-197: This conclusion cannot be drawn from the results that there was a 'sustained antibiotic pressure', so should be substantiated or rewritten.	This was a speculation based on the relationship between AMU and AMR. This has now been rewritten[line 294 page 10]
	Line 200-202: There is not always a direct link between the finding of resistance genes and use of an antimicrobial from the same class. Especially with 'MDR', there is often co-resistance (see e.g. Frontiers Co-resistance to Amoxicillin and Tetracycline as an Indicator of Multidrug Resistance in Escherichia coli Isolates From Animals). So the conclusion that a rise in tetQ and tetB suggests a higher use of tetracyclines cannot be drawn.	We have revised this to focus on the differential selection of specific resistance pathways. We continue to speculate on the use of tetracycline, as recent reports in Uganda indicate that this antibiotic is among the most commonly used in livestock. [Line 218, page 7]
	Line 204-211: See my previous remarks about 'transmission events' based on your classification. I doubt whether this is a sound method	We have made the following valid arguments as the basis of this analysis a) We assumed that sharing a multi-drug-resistant strain between a pair in a household is a rare

	to measure ‘transmission’ and should be better substantiated with good references. Even then, a common external source could be well possible.	event, but if it to happen I will most likely happed at farm level, and therefore, when detected represents a potential transmission event. So this analysis sought to demonstrate that the probability of an event will be higher at Farm level, then within a subcounty and least between districts/production settings. It is on the basis of this empirical observation that we make inferences on potential transmission events. We recognize the limitations of our coarse transmission unit(MDR), likely to introduce uncertainties. Thus, we refer to these as "potential transmission events" and recommend more granular genomic tools for future analysis.
	Line 212-216: The sentence does not read well; what is meant by “high carriage of the dfrA1 and tetQ genes relative to that of their pig”? Line 215 could be red as farmers using tmp/s themselves but this is not often used in humans as far as I know (but I’m not familiar with human use in Uganda).	This has now been revised on page 9 and line 254, as well as page 7 line 219 to add more clarity
	Line 224-228: I don’t support these conclusions. The only thing that can be concluded is that higher ABR rates and gene carriage (of selected antimicrobials) have been found on semi-intensive farming systems in both humans and animals. So this needs to be rewritten. Also the statement that the farm is the ideal level for interventions cannot be argued based on your findings and is also not discussed in the discussion part. Finally, I do not see how this approach with MDR-gene load relationships presents diagnostic opportunities for evaluations. It should be better substantiated and explained how you see this.	We have revised the conclusion as follows: In conclusion, our study highlights a significantly higher level of AMR and AMR gene carriage in peri-urban, semi-intensive pig production systems. Farmers in these systems carry more resistant sentinel bacteria compared to those in rural, free-range systems. Our findings suggest a potentially increased AMR risk in urban swine production in Uganda, which may intensify with growing demand for animal protein. Notably, our models indicate that farm-level factors account for most of the variation in AMR, suggesting that interventions should be focused at this level. [line 294 page 10]
	Materials and Methods This section is currently of poor scientific quality and clarity. A lot of crucial details are missing in the description of what has been done. Field study. It is not clear how monitored farms specifically were selected, how animals	

	were selected (only one animal per farm and same animal followed during the whole study?) and how samples were taken (e.g. swab, fresh faecal droppings). How was the ‘farmer’ defined (as especially in free-range systems there are mostly more people dealing with the animals)? It is also not mentioned how samples were stored and transported (e.g. transport conditions, transport time). It is also not clear how the reference group was selected (line 248-249); based on which criteria? How was defined whether someone was a ‘high’ or ‘low’ risk person?	
	Microbiological analyses. It should at least be summarized which laboratory procedures have been followed to isolate and identify E. coli and Klebsiella. How was finally one colony be selected for further analysis on ABR? Why was chosen to both include E. coli and Klebsiella as I might assume that E. coli can be found in every faecal sample. Disc diffusion was used, it should be clear from which company these discs were derived and which procedures were followed. Why were only these six antibiotic classes tested (and why both genta and streptomycin) and not others? How and when was ‘resistance’ measured? Which criteria where used to classify an isolate as ‘resistant’ (e.g. clinical breakpoints, ECOFFs, CLSI or EUCAST)? Why was chosen to look for gene carriage for only 4 selected genes? Why were these genes chosen and not others?	We have added a summarized laboratory procedure for identifying E. coli and Klebsiella. The choice of E. coli as a sentinel organism is elaborated in the section above. We have also clarified that disc diffusion was used to assess AMR, with interpretation based on EUCAST [line 349 page 11] guidelines(Version 2022). Additionally, we have included information on two aminoglycosides used in the livestock sector, noting that, due to their side effects in humans, they are reserved for escalated use. [line 343 page 11]
	Line 239: pairs were followed for 10 months I assume (as you have 6 time points and 12 months would require 7 time points).	To clarify, the Pairs were visited every 2 months, here the metadata collected was in reference to the past two weeks.

	Line 232: The National Council is from Uganda I suppose? Please make clear.	Yes indeed this is the Uganda National Council of Science and Technology
	Line 243-244: it should be clear that 35 pairs per study arm were followed (70 in total)	This has been clarified line 326 page 11
	Line 249: S1 refers to results so it should be referred to in the results section, not here.	This has now been revised
	Line 253: related to Table S1. Was this information requested to the farmers every visit (I assume only at the start which means that e.g. use of antibiotics is hard to analyze in a longitudinal study)? Why is the row 'antibiotic use' in referents almost empty? How to interpret 'antibiotic use' (Y/N) in the table referring to pigs? Is this antibiotic use in the last 2 weeks before the start of the study? Did it only refer to the animal that was sampled? Was it not monitored during each visit? Why was the type/class of antibiotic used not recorded? Figure S10: I don't really know how to interpret this figure, especially the numbers in the upper part (9 vs 10 samples of men and women? Where does this stand for?). And why is marital status important?	The questionnaire was administered on each visit. The antibiotic use response has now been edited accordingly The questions about antibiotic use, were in reference to the two weeks prior to each visit The detailed information about the antibiotics was not tracked to avoid making the questionnaire longer, which could increase non-responsiveness and attrition. The footnote for Figure S11 now clarifies the symbols: the brown, blue, and green icons represent high, low exposure references, and farmers, respectively. We also show how missing data arose using the gender variable.
	Line 261: seven antibiotics were tested, not six	Yes indeed they are 7 antibiotic, this has been rectified all through the manuscript
	Line 262-266: This assumption does not hold in my view and is also not really substantiated by Rhouma et al. I would like to see robust evidence for the assumption that AMR profiles in only one selected colony is indicative of AMR profiles in the whole population of Enterobacteriaceae in the gut. I think this is not true as there can be a wide variety of resistance profiles within a population.	The boom-and-bust phenomenon is well-established in microbial system dynamics and microbial ecology REF . A resource, such as the resistance ability of a bacterium, is exploited to dominate until that resource is out-competed. We assume that, numerically, it is more likely that when sampling, the dominant colony will be the one that is picked.
	Line 266-267: This is not clear. Did you extract an	This has now been clarified, an aliquot of extracted DNA was sent to Digital One Health Laboratory for down-

	aliquot of a fecal sample? Which methods were used? How was made sure that it contained ‘high molecular grade DNA’? Line 271: Tables S3&4 and Figure S4 should be referred to in the results section. In table S3, the legend talks about six tested antibiotics, but in fact it were 7. It is also not clear where the text on S page 9 (between table S3 and S4) about factors associated with ABR gene belongs to?	stream qPCR analysis of selected genes See page 11, line 342 The table as listed have now been revised see page 6, line 135,149,170,172) The number of tested antibiotics has now been rectified See page 11, line 349
	Statistical analyses. Line 302: What does time (T+1 mean? That is 2 months after the sampling moment in the farmer of that animal?	We aimed to examine whether a farmer's gene carriage influences the pig's gene carriage at the subsequent visit, and vice versa. In this context, if T represents the current time, then T+1 refers to the subsequent visit. See page 13, line 400
	Line 301-303: It is not clear what exactly was modelled. Only gene carriage (and then per each of the 4 different genes)?	Here we modelled AMR gene carriage, we have also modelled the genes individually and the results are similar See table 2 Model 1 and 4
	Line 303-308: It is not clear how ‘transmission’ was defined. You are assuming that sharing a multi-drug resistant strain between a pair is a rare event. But how was this analyzed? Was transmission suspected when isolates with exactly the same antimicrobial resistance profile (phenotypically) were found in both farmer and pig? Could it also be that this is not transmission but has a same external source (e.g. drinking water, environment)?	Here, the unit of transmission is defined as the sharing of an MDR pattern between the farmer and the pig. This sharing could occur at the same visit (Lag=0) or at different visits (Lag=1-3). The aim is to demonstrate that this sharing is more pronounced at the farmer level than between production systems. While some of this sharing may be due to external sources, the direct and indirect relationship between gene carriage (tetQ & ermB) in both the farmer and the pig suggests otherwise. That is to say it is more likely due to sharing than not See page 13, 402
	Figures and tables Many tables and figures must be improved. Often the legend does not clearly correspond with the figure and many figures are hard to interpret, even after reading the legend which means that the legend also should be rewritten to clearly explain the reader how to interpret the figures. For figures where correlations were calculated (e.g. 2b; 3e,	The tables and figures have now been fully edited to correspond with the tables.

	4b and others) I would suggest to draw a dot plot to see the individual observations rather than a straight line indicating the relationship.	

Query	Reviewer comments	Author Response
Reviewer #1	L 41 To refer to shared resistance (phenotypic or genes) as transmission or transmission events is misleading and not scientific sound. Please use the wording potential transmission events throughout the MS and not only in the discussion.	This has now been edited on L39 and all throughout the manuscript as advised
	L304 missing the reference	This has now been added see Line 3
	L 308-309, isn't "interaction between" more precise than "influence"	This has been clarified in line 190—it is the interaction between production system and geography that was not addressed in this study. Line 406 has also been revised accordingly.
	L 482 "Livestock use"?	
	L 488 AMR and MDR aren't two different entities justifying "and", please reword.	This has been reworded as requested on Line 274
	L 489 (b) this statement is confusing – please be more precise	This has been revised to improve clarity from line 274 to 276
Reviewer #2	-Carefully check all the grammar/spelling and the insertion of proper references	Tizian and Paul please read through for a final refinement .
	-Change ABU into AMU where applicable (also in figures)	This has been revised at Line 335, as well as Figure 5
Reviewer #3	Line 46 - 47: The OR and stats outcomes of the last item (copy number of tetQ) is missing in the abstract.	These stats were not included in the abstract due to word limits. This is however elaborated in the results and discussion section.
	For my question regarding average number of pigs and treatment protocol, I was considering the farm characteristics rather than farmers', which is being summarized in Table S1. Does the data about the number of pigs that each farmer was raising and whether farmers had a protocol to follow for AMU available?	The data in Table S1 is limited to information recorded for the sow included in this study. Farmers were unwilling to disclose the total number of animals due to concerns about potential tax implications; as a result, this information was not collected.